

**Variability in grain size, mineralogy, and mode of occurrence of Fe in surface**
**sediments of preferential dust-source inland drainage basins: The case of the**
**Lower Drâa Valley, S Morocco**
Adolfo González-Romero[1,2,3], Cristina González-Florez[1,3], Agnesh Panta[4], Jesús Yus-Díez[2,a], Cristina Reche[2],
Patricia Córdoba[2], Andres Alastuey[2], Konrad Kandler[4], Martina Klose[5], Clarissa Baldo[6], Roger N. Clark[7],
Zong Bo Shi[6], Xavier Querol[2], Carlos Pérez García-Pando[1,8]
[1]Barcelona Supercomputing Center (BSC), Barcelona, Spain
[2]Spanish Research Council, Institute of Environmental Assessment and water Research (IDAEA-CSIC),
Barcelona, Spain
[3]Polytechnical University of Catalonia (UPC), environmental engineering doctoral programme, Barcelona,
Spain
[4]Institute of Applied Geosciences, Technical University Darmstadt, Darmstadt, Germany
[5]Karlsruhe Institute of Technology (KIT), Institute of Meteorology and Climate Research (IMK-TRO),
Department Troposphere Research, Karlsruhe, Germany
[6]School of Geography Earth and Environmental Sciences, the University of Birmingham, Birmingham,
United Kingdom
[7]PSI Planetary Science Institute, Tucson, AZ, USA
[8]Catalan Institution for Research and Advanced Studies (ICREA), Barcelona, Spain
[a]now at: Center for Atmospheric Research, University of Nova Gorica, Ajdovščina, Slovenia.
**Corresponding author:**
Adolfo González-Romero, <agonzal3@bsc.es>
Xavier Querol Carceller, <xavier.querol@idaea.csic.es>



**Abstract**

The effect of mineral dust emitted from arid and semiarid surfaces upon climate and ecosystems depends fundamentally on their particle size distribution (PSD) and size-resolved mineralogical composition. However, soil mineralogy atlases used for mineral-speciated dust modelling are highly uncertain as they are derived extrapolating mineralogical analyses of soil samples that are particularly scarce in dust-source regions. This extrapolation neglects the processes affecting the formation of different dust-emitting surface sediments, such as dunes, crusts, and paved sediments. The Lower Drâa Valley, an inland drainage basin and preferential dust-source located in southern Morocco, was chosen for a comprehensive analysis of sediment grain size and mineralogy. Different sediment types samples were collected, including paleo-sediments, paved surfaces, crusts, and dunes, and analysed through PSD analysis of minimally and fully dispersed samples, and X-ray diffraction mineralogical analysis of bulk samples. We also performed Fe sequential wet extraction to characterize Fe mineralogy, including the contents of (oxyhydr)oxides (goethite and hematite), key to dust radiative effects, and poorly crystalline pool of Fe (readily exchangeable ionic Fe and nano-Fe-oxides), relevant to dust impacts upon ocean biogeochemistry. Based on the results we propose a conceptual model where both particle size and mineralogy are segregated by transport and deposition of sediments during runoff of water across the basin, and by the precipitation of salts, which causes a sedimentary fractionation. Coarser particles substantially richer in quartz are more present in elevated areas, and finer particles rich in clay, carbonates, and Fe-oxides are present in depressed areas, where dust emission is maximized. When water ponds and evaporates, secondary carbonates and salts precipitate, and the clays are enriched in readily exchangeable ionic Fe, due to sorption of dissolved Fe by illite. Our results differ from currently available mineralogical atlases and highlight the need for observationally-constrained global high-resolution mineralogical data for mineral-speciated dust modeling.

**Keywords:** Arid regions, dust-sources, desert dust, dust-emitting sediments formation model, dust modelling.



**1. Introduction**

Desert dust is atmospheric particulate matter (PM), mostly mineral in composition, emitted into the atmosphere by wind erosion of arid and semi-arid surfaces. The global dust source regions include North Africa, the Middle East, Central Asia, Western Australia, South America, Southern Africa and Southern US-Northern Mexico. From these regions, North Africa accounts for around 50 % of the global dust emissions, followed by Central Asia, the Middle East and East Asia (Kok et al., 2021). Dust storms arise when strong winds generate a large amount of dust particles that drastically reduce visibility nearby and are transported over distances of hundreds of kilometres (Prospero et al. 2002). During transport, dust perturbs the energy and water cycles by direct radiative forcing and influences cloud formation, precipitation and the associated indirect radiative forcing (Weaver et al., 2002). Dust transports nutrients across the planet affecting ocean productivity (Boyd et al., 2007), plant nutrient gain or loss (Sullivan et al., 2007), and glacier mass budgets (Goudie & Middleton, 2006). Dust can also directly affect human health by inhalation or by favouring the propagation of diseases (Goudie & Middleton, 2006, De Longeville et al., 2010; Karanasiou et al., 2012; Pérez García-Pando et al., 2014). It can reduce renewable solar energy output due to attenuation of solar radiation and soiling of solar panels (Monteiro et al., 2022), create poor visibility on roads increasing the risk of traffic accidents (Middleton, 2017) and cause disturbances in airport operations and air traffic (Monteiro et al., 2022).

Dust is emitted mostly from arid inland drainage basins (Dubief, 1977; Prospero et al., 2002; Goudie & Middleton, 2006; Bullard et al., 2011; Querol et al., 2019). These basins encompass different sedimentary environments, many of which are potentially efficient sources of dust, including unconsolidated aeolian deposits, endorheic depressions, and fluvial and alluvial dominated systems (Bullard et al., 2011). Consolidated or compacted fine sediments in the form of crusts and paved sediments, for instance on ephemeral lake beds, can also be important dust emitting surfaces when loose sand size sediments provided by adjacent sand dunes are available (Stout, 2003). These sand particles are efficiently mobilised by wind and strike the consolidated surface breaking the sediment aggregates and releasing dust (Shao et al., 2011).

Models developed to simulate the atmospheric dust cycle and its impact on climate represent dust emission, transport, interactions with radiation and clouds, and removal by wet and dry deposition (Tegen and Fung, 1994; Ginoux et al., 2001; Zender et al., 2004; Perez et al., 2011, Klose et al., 2021). Modelling efforts have mostly focused on the representation of dust sources and emission (Kok et al., 2021) and the characterization of dust sources is one of the crucial aspects for representing dust mobilisation in models. Traditionally, models used aridity as a criterion to identify potential dust sources (Tegen and Fung, 1994). Satellite retrievals subsequently showed that the most prolific sources occupy a small fraction of arid regions (Prospero et al., 2002; Ginoux et al., 2012). These so-called "preferential sources" are found within enclosed basins, where easily eroded soil particles accumulate after fluvial erosion of the surrounding high-lands. The implementation of preferential source functions in global models based on topography (Ginoux et al., 2001), hydrology (Tegen et al. 2002; Zender et al. 2003), geomorphology (Bullard et al., 2011), or satellite proxies (Prospero et al., 2002; Gioux et al., 2012), has significantly improved the skill of models by approximately locating large-scale natural sources. However, models are not able yet to capture the small-scale spatial and temporal variability in emissions apparent from observations. Some studies have provided small-scale understanding on the role of geomorphology and sedimentology upon dust



emissions (Bullard et al., 2011; Baddock et al., 2016). For instance, Bullard et al. (2011)
developed a conceptual model of how different geomorphologic surfaces affect the intensity
and temporal variability in dust emissions.
While it is key to understand dust source location and emission intensity, climate impacts by
dust also depend upon its mineralogy. Dust is a mixture of different minerals including quartz,
clay minerals (mica/illite, kaolinite, palygorskite, chlorite/clinochlore and
smectite/montmorillonite), feldspars (albite/anorthite and orthoclase), carbonate minerals
(mainly calcite and dolomite), salts (mainly halite and gypsum), Fe-oxides and hydroxides
(mostly goethite and hematite) and other oxides or hydroxides of Ti, Mn and Al (Coudé-Gaussen
et al., 1987; Schültz & Sebert, 1987; Molinaroli et al. 1993; Gomes, 1990; Sabre, 1997;
Caquineau, 1997; Avila et al., 1997; Caquineau et al, 1998; Claquin et al., 1999; Formenti et al.,
2008; Nickovic et al., 2012; Scheuvens et al., 2013; Journet et al., 2014; Scanza et al., 2015; Ito
& Wagai, 2017; Querol et al., 2019). The relative abundances, size, shape, and mixing state of
these minerals influence the effect of dust upon climate. For instance, the absorption of solar
radiation by dust depends upon the iron oxide content (Tegen et al., 1997; Sokolik and Toon,
1999; Reynolds et al., 2014, Di Biagio et al., 2019), ice nucleation in mixed-phase clouds is highly
sensitive to the amount of K-feldspar and quartz (Boose et al., 2016b; Harrison et al., 2019), and
the bioavailability of iron in dust depends upon its iron mineralogy and speciation (Shi et al.,
2012). According to the geological, geomorphological and climate (weathering) patterns of the
desert regions, the type, and proportions of minerals might greatly vary (Caquineau, 1997;
Caquineau et al, 1998, Claquin et al., 1999; among others). For example, Sahelian dust is
composed mainly of quartz, kaolinite and hematite, while in North-eastern China and the Sahara
mica/illite, kaolinite, quartz and carbonates prevail (Shen et al., 2009; Claquin et al., 1999).
Despite the potential importance of dust mineralogical variations, climate models typically
assume dust composition as globally uniform, which is partly due to our limited knowledge of
the composition of the parent sources at global scale. The few models that explicitly represent
dust mineralogical composition (e.g., Scanza et al., 2015; Perlwitz et al., 2015, Li et al., 2021;
Gonçalves Ageitos et al. 2023) use global atlases of soil type and the relation of this variable to
soil mineralogy. This relation is inferred using massive extrapolation from a limited amount of
mineralogical analyses, particularly in dust source regions, ancillary information on soil texture
and colour, and a number of additional assumptions (Claquin et al., 1999; Journet et al., 2014).
The mineralogical composition is characterised in two traditional grain-size ranges (Wentworth
(1922) and Urquhart (1959)), i.e. clay (<2 µm) and silt (2-63 µm) linked to FAO (Food and
Agricultural Organization of the United States) soil texture datasets based on measurements
following wet sieving, a technique that disperses (breaks up) the mineral aggregates found in
the undisturbed parent soil into smaller particles (Chatenet et al., 1996). Furthermore, the
samples that underpin these atlases consider the first 10-15 cm of soil sediment, which is much
deeper than the thin layer that is relevant to wind erosion and dust emission, and mineralogy is
normally analysed after removing organic matter with hydrogen peroxide ($H_2O_2$), which can
partially dissolve carbonate minerals.
The assumed relationship between mineralogy and soil type in these atlases neglects the role of
geomorphology and sedimentology affecting the formation of different dust-emitting surface
sediments, such as dunes, crusts, and paved sediments. In this study, we provide a
comprehensive analysis of the variability in grain size, mineralogical composition and Fe





mineralogy and speciation of sediments collected across the Lower Drâa Valley, an inland
drainage basin and prolific dust-source located in the north-western border of the Saharan
desert in southern Morocco (Figure 1). The data collection was performed during a wind erosion
and dust field campaign in September 2019 in the context of the FRontiers in dust minerAloGical
coMposition and its Effects upoN climaTe (FRAGMENT) project. Based on the analysis of the
results we propose a conceptual model that links formation processes of potential dust-emitting
sediments to their particle size distribution (PSD) and mineralogy across the basin.

## 2. Methodology

### 2.1 The FRAGMENT field campaign and the study area

The sediment samples analysed in this study were collected during a field campaign that took
place in September 2019 in the Lower Drâa Valley, west of M'Hamid, between the Erg Chigaga
and L'Bour (Figure 1a), a dry inland drainage basin where dust emission is frequent as evidenced
by satellite data (Ginoux et al. 2012) (Figure 1b). The region lies where the Sahara Desert begins,
to the south of the Atlas Mountain, near the Algerian border, in the Drâa River Basin. Preliminary
results from the Earth Surface Mineral Dust Source Investigation, EMIT, (Green et al., 2020) show
the presence of a complex regional mineralogy with fine-grained goethite, hematite (with
substantial nano-sized hematite), gypsum sulphate salts in the lowlands (depressions) and
Illite/muscovite, with local outcrops of carbonates in the study area (Figure 1c). The EMIT
mineral maps show that the study area is representative of the larger area.
The campaign was conducted in the framework of the FRAGMENT project, in which distinct
desert dust source regions are being characterised to better understand the size-resolved dust
emission and composition for different meteorological and soil conditions. The aim of
FRAGMENT is to better understand dust emission, its mineralogical composition and the effects
of dust upon climate, by combining field measurements, laboratory analyses, remote and in situ
spectroscopy, theory and modelling. FRAGMENT field campaigns consist of intensive sediment
sampling and meteorological and airborne dust measurements in one specific location, along
with sediment sampling across the broader basin. The intensive meteorological and airborne
dust measurements were performed in the dry lake L'Bour and are analysed in e. g., González-
Florez et al., 2022; Panta et al., 2022; Yus et al., in prep. Here we focus on the sediment sampling
across the basin.
The study area records very low annual precipitation (ranging from <50 to 800 mm) and
extremely variable droughts interrupted by extreme floodings (Berger et al., 2021). The Drâa
River was anthropogenically dried in this area mostly due to the construction of El Mansour
Eddahbi dam in 1972 (near Ouarzazate). The Jbel Hassan Brahim range reaches the highest
altitude in the area (840 m.a.s.l.), while the Drâa River is the lowest point (570 m.a.s.l.). The
study region corresponds to a low relief alluvial system, unarmored and unincised according to
Bullard et al. (2011). Rains are scarce, but sometimes they concentrate in the mountains (high-
lands) and even more sporadically they can directly affect the area during convective storms,
creating flash floods with a high sediment load canalised by torrents or wadis, such as wadi
Latache (high-lands) (Figure 1a), which flood flat areas. In specific areas across the basin, highly
sediment-loaded waters can be shortly ponded on the way to Drâa River in small depressions,
such as dry Lake Iriki, Erg Smar or L'Bour (low-lands) (Figure 1a), among other areas, along the
floodplain. Dunes are concentrated in small flat areas, near depressions, where, after wind
erosion, sediment can be dragged and be entrapped by the very scarce and low vegetation.



**2.2 Sediment sampling**


The sampled sediments include paleo-sediments (hereafter named sediments), paved
sediments, crusts, and dunes, according to the classification by Watt & Valentin (1992) and
Valentin & Bresson (1992). Paved sediments result from cyclic drying and aeolian erosion of the
surface of paleo-sediments and range from 0.5 to 2 cm of vertical depth. Crusts ranged from 0.1
to 2 cm of vertical depth and we differentiated two types: i) thin depositional crusts formed as
result of the deposition of sediments from running water during floods, and ii) thicker
sedimentation crusts resulting from the sedimentation and drying of highly sediment-loaded
waters in ponded areas of different sizes. Sediments are below the crusts (not exposed to the
atmosphere) and dunes are aeolian deposits. We used a 50 cm$^2$ inox steel shovel to sample
surfaces (first top cm), sediments (below surface, from 1 to 5 cm in the vertical depth) and dunes
(from surface to 5 cm). We registered coordinates, type of sample, surroundings description and
we also recorded any other important information and made concept drawings. We obtained
sediment samples, including crusts (12), dunes (12), paved sediments (11) and sediments (7)
(Figure 2) from different locations in the Drâa River Basin.

**2.3 Sample treatment**


To analyse mineral-size fractionation (<10 and 10-63 µm) we applied a fully dispersed size
fractionation using MilliQ-grade water and shaking the samples previous to separation for 12-
24 h. First, samples were subjected to 250, 63 and 10 µm sieves to obtain the <63 and <10 µm
fractions. Due to availability, the smallest opening size of the sieve was 10 µm. Sonic sieving was
applied for 60 s at maximum sustainable power for 3 min in every sieve. Finally, subsequent
drying at 80 °C was applied to recover the solid fraction from the suspension.

**2.4 Analysis**


2.4.1 Particle size distribution and texture
The particle size analysis was carried out for fully (natural aggregates totally dispersed) and
minimally (natural aggregates minimally dispersed) dispersed PSD to obtain the fully dispersed
particle size distribution (FDPSD) and the minimally dispersed particle size distribution (MDPSD)
to evaluate (i) how aggregates and particles occur in natural conditions (MDPSD) and (ii) the
distribution of single particles that form the aggregates (FDPSD). The MDPSDs were obtained
with laser diffraction using a Malvern Mastersizer 2000 Scirocco accessory (hereinafter,
Scirocco) for minimally dispersed conditions. In this case, samples of 0.3-0.5 g of the fraction <2
mm were introduced into the Scirocco vibration plate with a 2 mm aperture and 5 s measuring
time. FDPSDs were determined using the Malvern Mastersizer 2000 Hydro G accessory
(hereinafter, Hydro) with a water suspension and ultrasound assistance for totally dispersed
conditions. In that case, the samples were pre-treated following the method by Sperazza et al.
(2004). The suspension was introduced into the Hydro's sample container, pumping at 1750 rpm
and stirring at 500 rpm. Results were obtained in both cases using the Fraunhofer method (Etzler
et al., 1997).
To investigate the possible occurrence of vertical segregation of the PSD (top layers are the ones
that are emitting dust), 7 crust and 5 paved sediment samples were selected for vertically-
resolved PSD analyses. To this end, 3 sub-samples were extracted from each sample (top,



middle, and bottom sections) by scratching the surface with a cutter from top to bottom and
were analysed separately. The thickness of these crusts varied between 4 to 8 mm.
The pipette method was also used to analyse the texture of a soil layer or boundary according
to FAO-UNESCO (1990) of a total of six samples. This allows us to separate a suspension of the
sample in MilliQ-grade water into different size fractions (>63, 2-63 and <2 µm), dry and analyse
each size-fraction individually.
2.4.2 Mineralogical composition
Quantitative mineralogical analyses of bulk sediment samples and segregated size fractions
were carried out by means of powder X-Ray Diffraction (XRD), using a Bruker D8 A25 Advance,
with Cu K$_{\alpha 1}$ radiation of 1.5405 Å wavelength, a Bragg Brentano geometry and a LynxEyeXE
detector. Analysis was performed at 40 mA and 40 kV with a range of angles from 4 to 60° and
angle steps of 0.019° and 10 Hz for 1 h/sample. The mineral identification was made with the
EVA software package by Bruker. For quantitative analyses we used the method of the internal
reference material by Chung (1974), with quartz as the internal reference. The ratios of
intensities of the different minerals versus quartz were obtained by preparing and analysing
binary mixtures of the specific minerals and quartz. The accuracy of the XRD quantitative
approach was tested by analysing 16 mixtures of reference materials with known concentrations
of minerals. Figure S1 summarises major results, which yield relative standard deviations versus
the known contents of quartz (13 % of error), albite/anorthite (10 %), calcite (31 %), dolomite
(14 %), mica/illite (29 %), kaolinite (11 %), gypsum (27 %), anhydrite (19 %), goethite (42 %),
hematite (50 %).
For an in-depth evaluation of clay mineralogy, XRD analyses of oriented aggregates following
the procedure by Thorez (1976) were carried out for the same six samples of the texture. We
treat the samples for air drying (AO), glycolation (AG) and heating (AC). Mica/illite,
chlorite/kaolinite, palygorskite and smectite were found in all the samples, as evidenced from
the bulk XRD analysis. Calcite and dolomite were dissolved by acidifying soil suspension with a
strong acid as HCl and the excess used to quantify stoichiometrically the content of carbonates
using the method proposed by Horváth et al. (2005) also for the same six samples of the clay-
oriented aggregates and texture.
To investigate the possible occurrence of mineralogical vertical segregation, the 7 crust and 5
paved sediment unaltered samples used for particle size analysis (see section 2.4.1) were also
used for vertically-resolved mineralogy analyses.
2.4.3 Mode of occurrence of Fe
Fe is a key ingredient to climatic and biological processes affected by dust. For instance, the
amount, mixing state and size of Fe-oxy/hydroxides determine the degree of absorption of solar
radiation by dust (Engelbrecht et al., 2016) and the potential solubility of the dust deposited into
the ocean (Shi et al., 2012). However, the XRD semiquantitative analysis for Fe-oxy/hydroxides
are affected by large uncertainties due to the low concentrations (increasing relative errors) and
is not sensitive to nano-Fe-oxides (Shi et al., 2012). We complemented the XRD analyses by
quantifying the levels and mode of occurrence of Fe in the bulk samples using the methodology
described in Shi et al. (2009), through which based on a sequential extraction we determine the
amount of readily exchangeable (adsorbed) Fe ions and nano-Fe-oxides (FeA) and the amount



of crystalline Fe-oxides, mainly hematite and goethite (FeD) in the samples. We used 30 mg of
Arizona Test Dust (ATD; ISO 12103-1, A1 Ultrafine Test Dust; Powder Technology Inc.) to test the
accuracy of the method and extractions were done with 15 ml of extractant solution. For total
Fe content (FeT) we used a two-step wet acid digestion method developed by Querol et al.
(1993, 1997) and a coal fly ash (1633b) standard sample was used to test accuracy. The 1633b
gave 7.5 % with a standard deviation of 0.14 % for total Fe (reference content of 7.8 % of Fe),
while ATD gave 0.076 % with a standard deviation of 0.002 % of FeA and 0.49 % with a standard
deviation of 0.07 % for FeD + FeA (reference content of 0.067 % of FeA and 0.41 % of FeD).
Furthermore, by subtraction, we obtained the contents of structural Fe (FeS = FeT - (FeA + FeD)),
corresponding to the Fe fraction as elemental Fe into the structure of minerals other than Fe-
oxides, such as illite or other Fe-bearing minerals. Furthermore, the FeD contents were
converted stoichiometrically to hematite ($Fe_2O_3$) and goethite (FeO(OH)) by using the
hematite/goethite proportions from XRD.
2.4.5 Electron microscopy of crust and paved sediment sections
The PSD, mineralogy and morphology of crust and paved sediments can vary along the vertical
profile, especially in crusts where progressive sedimentation and subsequent evaporation leads
to inter-layering of sediments with different properties. For that purpose, crust and paved
sediment sections were impregnated with epoxy resin, cut, and polished with diamond paste
for microscopy analysis. The polished samples were coated with graphite before analysis with a
JEOM JSM-7001F SEM-EDX Scanning Electron Microscope (SEM).
**3.  Results and discussion**
**3.1 Regional variability**
3.1.1 Particle size distribution
We analyse the PSDs of the samples collected across the basin to detect possible trends or size
segregation patterns from high- to low-lands for the different types of sediment. The mean
median diameter values of each group of sediments provided in this section represent the mean
and standard deviation of the median diameters. Because the PSDs are generally bi-modal, other
PSD metrics can be found in Table 1, including the maximum, minimum and mean of the median
diameters for different types of sediments, location, PSD type (MDPSD and FDPS), and size
fraction (full range, <63 µm and >63 to < 2000 µm).
MDPSDs, excluding dune samples, show a major mode centred around 100 µm in diameter and
a secondary one between 2 to 20 µm (Figure 3a; Table 1). FDPSD's also show two modes at 5
and 100 µm (Figure 3b; Table 1). The MDPSDs and FDPSDs of dune samples are very similar with
a main mode centred around 150 µm and a secondary small one at 5 µm (30 times lower) (Figure
3c and d). Crust samples show the largest fine (0-5 µm) fraction in MDPSD, followed by paved
sediments and sediments (Figure 3e). FDPSDs show a similar trend but with a larger proportion
of fine particles compared to MDPSD (Figure 3f).
The mean median diameter of the MDPSDs (Figure 4a), excluding dune samples, is 88±63 µm;
and that of the FDPSDs, is 27±51 µm (Figure 4a). Therefore, aggregates are about 3 times coarser
than individual mineral particles. As expected, dunes were coarser than other types of
sediments, with a mean median diameter of 219±70 µm of the FDPSDs, which is very similar to



that of the MDPSDs (Figure 4b). The mean median diameters of MDPSDs are 70±48, 74±45 and
113±79 µm for sediments, paved sediments and crusts, respectively (Figure 4c); whereas the
mean diameters of FDPSDs are 19±11, 21±26 and 37±77 µm for sediments, paved sediments
and crusts respectively, about 3 to 4 times finer (Figure 4d).
The spatial variation of the mean diameter of the FDPSDs (Figure 5) shows coarser crusts (>40
µm) close to the high-land areas, and finer crusts (<40 µm) near the Drâa River, likely due to
flooding (causing transport and deposition of fine sediments, especially in the low-lands) caused
during scarce and intensive rains. For paved sediments, sediments and dunes, spatial PSD trends
were not evident, with mean median diameters ranging from 10 to 120, 10 to 40 and 120 to 300
µm, respectively, randomly located across the basin (Table 1).
According to the size classification by Valentin & Bresson (1992) and using the FDPSD data
(Figure S2), dune samples can be classified as sand, loamy sand, and sandy loam; sediments as
silt loam and loam; paved sediments as sandy loam, loam and silt loam; and crusts as sandy
loam, loam, silty clay loam and silt loam. As shown in Figure S2 and due the higher transport
potential of clays during rain episodes, and their accumulation during ponding, crusts tend to be
further enriched in clay fractions, especially in low-lands, compared to paved sediments and
sediment samples (see section 3.4).
3.1.2 Mineralogical composition
We describe here the mineralogy of samples collected across the basin to detect possible trends
or mineral segregation patterns from high- to low-lands for the different types of sediment. The
mineralogical composition (mass % composition of the bulk sample) of dunes, crusts, paved
sediments and sediment samples is summarised in Table 2. Dunes show a homogeneous
mineralogy across the study area, with mineral abundances of 74±9.7 % quartz, 5.8±2.9 %
calcite, 6.7±3.6 % microcline, 6.9±3.1 % albite/anorthite, 4.1±2.3 % clay minerals, 1.0±1.4 %
dolomite, 0.38±0.26 % goethite and 0.12±0.11 % hematite and trace amounts of halite and
gypsum (<0.1 %) (Figure 6).  In comparison to dunes, crusts are depleted in quartz (48±11 %) and
feldspars (5.0±2.1 % albite/anorthite and 4.4±3.1 % microcline), and enriched in clay minerals
(17±8.0 %), calcite (19±8.0 %), dolomite (3.0±1.3 %) and Fe-oxides (0.24±0.28 % hematite and
0.42±0.56 % goethite) (Figure 6). The content of gypsum (0.23±0.56 %) and halite (2.9±5.1 %) is
higher than in dune samples, but variability is large because it depends on the exact point of
crust sampling. Paved sediments have a similar mineralogy than crusts, for quartz (51±8.7 %),
calcite (17±4.9 %), clay minerals (16±7.3 %), albite/anorthite (6.3±1.8 %), microcline (4.5±2.5 %),
dolomite (3.5±0.79 %), hematite (0.34±0.25 %), and goethite (0.38±0.38 %), but with lower
content of gypsum (<0.1 %) (Figure 6). Sediments are also similar to paved sediments and crusts
with a mean quartz content (55±11 %), calcite (17±4.6 %), clay minerals (14±6.8 %),
albite/anorthite (5.8±1.5 %), microcline (3.7±1.6 %), dolomite (3.4±1.3 %), hematite (0.28±0.37
%) and goethite (0.37±0.32 %). Trace amounts of gypsum (<0.1 %) and halite (0.32±0.55 %) were
also found in sediments (Figure 6).
In comparison with the bulk sediment, the fully dispersed silt fraction (10-63 µm) shows a lower
amount of quartz (35±6.4 %) and feldspars (7.4±2.5 %), a higher content of carbonates (25±5.2
%), clays (22±10 %) and hematite (1.07±0.38 %) and a similar content of goethite (0.61±0.32 %).
In the fully dispersed <10 µm sieved fraction, the amount of quartz (23±5.2 %) and feldspars
(4.7±1.1 %) is two times lower than in the bulk sediments. The fraction of carbonates remains



similar (21±9.0 %) and the content of clays increases substantially (38±9.8 %) compared to the
bulk and silt-size mineralogy. The Fe-oxide content increases by about a factor two for both
hematite (2.2±2.0 %) and goethite (1.8±1.2 %). Table 3 compares our mineralogical results in the
clay and silt size ranges, both with the fully dispersed separation and the pipette methods,
against the corresponding values provided by the available global mineralogical atlases of
Claquin et al. (1999) and Journet et al. (2014), which assume our sample locations to be either
fluvisols or yermosols in terms of soil type. In the silt-size fraction, we find similar contents of
quartz, total clay, mica/illite, chlorite+kaolinite, calcite and Fe-oxides, but 3 times less feldspars
and 5 times more dolomite. Compared to the clay-size fraction in the atlases, our <10 µm
fraction, shows larger content of quartz and feldspars (by factor of 2 to 4), a 30 % lower total
clay content and similar contents of calcite and Fe-oxides, which can only be partly explained by
the difference in the size fraction considered (<10 µm vs <2 µm) as shown by the results obtained
with the pipette method. Because kaolinite and chlorite have coincident spacing at 7 Å in the
XRD spectra, in current atlases these minerals may be confounded, whereas in our study we
quantified chlorite separately by identifying other minor peaks in the spectra. This is relevant as
both minerals are very different in terms of chemical composition. In our study, we also detected
minor concentrations of dolomite and traces of smectite and palygorskite. The large differences
in the silt-size feldspar content may be largely due to the lack of data and coarse assumptions
used in current atlases.
In our analysis of trends in mineralogy from the high-lands to the low-lands we considered all
sample types except dunes. The low-lands, such as L'Bour and Erg Smar, are enriched in clay
minerals (17±9.6 and 14±3.4 %, respectively) compared to the high-lands (9.1±0.97 %) (Figure
6). Mica/illite is the most common clay mineral reaching mean contents of 9.1±4.8, 8.1±2.0 in
Erg Smar and L'Bour, respectively, and 5.0±0.70 % in the high-lands. Kaolinite reaches 7.2±5.4,
4.9±2.1 and 3.5±0.30 % and clinochlore 1.7±1.8, 1.3±0.67 and 0.49±0.38 %, respectively.
Smectite and palygorskite were detected only in trace amounts (<0.1 %) in most samples, with
only palygorskite at Erg Smar and high-lands reaching a mean content of 0.34±0.58 and
0.15±0.06 %, respectively. The same trend is found for calcite (24±13, 16±3.1 and 11±2.7 %, Erg
Smar, L'Bour and high-lands), dolomite (5.0±5.1, 3.6±0.51 and 1.7±0.50 %, at Erg Smar, L'Bour
and high-lands) and Fe-oxides (0.78±1.4, 0.37±0.43 and 0.08±0.04 % for hematite at Erg Smar,
L'Bour and high-lands and 0.42±0.51, 0.39±0.35 and 0.32±0.21 % for goethite at Erg Smar, L'Bour
and high-lands) being steeper for hematite than goethite (Figure 6). Quartz follows an opposite
trend, increasing towards the high-lands (42±18, 53±5.0 and 61±5.4 %, at Erg Smar, L'Bour and
high-lands, respectively) (Figure 6). Albite/anorthite and microcline do not show a clear trend,
with 5.5±2.3, 5.9±1.8 and 5.4±1.2 % at Erg Smar, L'Bour and high-lands, and 3.4±2.4, 5.0±3.4 and
4.6±1.7 %, respectively (Figure 6). Salt concentrations peak randomly and depend on very local
scale conditions, being higher at concave areas where ponding is favoured (see section 3.4). The
mean content of halite is 1.0±2.2, <0.1 and 4.0±7.7 % at Erg Smar, L'Bour and high-lands and
that of gypsum is 0.18±0.35, <0.1 and 0.15±0.92 %, respectively (Figure 6).
A soft crust occurred on the surface of several dunes (Figure 2). The PSD and mineralogical
analysis of the crust and the underlying sands did not reveal significant differences. Pye & Tsoar
(2015) reported that surface hardening of dunes is due to the scavenging and deposition of clays
from suspended dust in light rains and by cementation of sand grains (meniscus) by precipitation

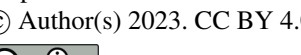


of carbonates and silica in the retained interstitial pore water. In both cases the potential
variability caused by the slight increase of this clay and carbonate/silica cementation is obscured
by variations in the bulk mineralogy.

**3.2 Vertical segregation in crust and paved sediments**

The examination of thin vertical cross-sections provides insight into how particle size and
mineral composition vary within the top few μm or mm of the surface. These differences are
relevant to the mineralogy and PSD of newly emitted dust.
The MDPSDs of crust sections (top, middle and bottom) are very similar, with two modes of
occurrence at 5-7 and 200 μm (Figure S3a). Yet, while the FDPSDs show similar two modes at 1-
5 and 100 μm for the top and middle sections and a second mode at 300 μm for the bottom
section (Figure S3b). The MDPSD mean median diameter of the 7 crust profiles reach 25±25,
54±80 and 25±26 μm for the top, middle and bottom sections, respectively, while FDPSD means
are 9.4±9.4 and 11±9.5 μm in the top and middle sections and 94±145 μm in the bottom one
(Figure S3c and d). Therefore, during the initial stages of ponding, coarser particles are deposited
first while finer particles remain suspended (see section 3.4) in the later stages before
evaporation of the water. Even some oxides, carbonates and salts may precipitate in the top
layers of the crust as water evaporates and the ionic strength increases.
No vertical PSD segregation is observed in paved sediments, but some top sections analysed
show enrichment in coarser fractions in FDPSD (the median diameter increases from bottom
and middle sections (14±6.8 and 12±5.8 μm) to the top section (23±28 μm)), likely due to
preferential erosion of finer fractions through sandblasting (see section 3.4).
The mean levels of quartz and feldspars are enriched in the bottom sections of the crusts (46±17
and 8.7±4.6 %, respectively) compared to the middle (38±11 and 8.3±2.5 %) and top sections
(41±12 and 6.9±2.2 %) due to the higher quartz content of the coarse fraction that is deposited
first (see section 3.4). The content of clay minerals, salts and Fe-oxides is similar in the top
(20±7.2, <0.1 and 3.3±1.9 %, respectively), middle (21±5.0, <0.1 and 2.8±1.6 %), and bottom
sections (19±9.1, <0.1 and 1.9±1.0 %). Carbonate minerals are relatively homogeneous, but
slightly enriched in the middle and top sections (29±9.7 and 28±7.9 %, respectively) compared
to the bottom section (24±8.4 %). This can arise from both detrital carbonate particles and
precipitation from high ionic strength waters that are ponded and dried in the low-lands.
The mineral composition of the paved sediment profiles differs slightly from that of crust
profiles. This is because the latter are affected by particle segregation during transport and
subsequent sedimentation. The top section of the paved sediment profiles has more quartz than
in the middle and bottom sections (44±8.1, 38±5.7 and 40±9.8 %, respectively), whereas
feldspars decrease from the bottom and middle to the top sections (9.1±4.2, 9.3±2.2 and 6.9±2.7
%). Carbonates and clay are relatively homogeneous (26±4.9, 26±2.0 and 25±4.2 % for
carbonates, and 22±8.4, 23±9.2 and 25±4.9 % for clays, respectively). The slight depletion of
minerals in the top section may be due to sandblasting, which tends to erode the fine fraction
of the surface over time (see section 3.4). Fe-oxides are more present in the top section than in
the middle and bottom sections (2.1±0.47, 2.0±0.38 and 1.7±0.27 %, respectively) and the
presence of salts is very low (<0.1 % for all sections).





**3.3 Mode of occurrence of Fe**
We implemented a sequential Fe extraction procedure to evaluate the levels and mode of
occurrence of Fe in dust samples from the basin. Due to limitations of XRD analysis for low Fe-
oxide contents, this procedure provided a much more precise quantitative evaluation.
The mean FeT content of bulk crusts, paved sediments and sediments was found to be 3.6±0.71,
3.4±0.47, and 3.2±0.44 %, respectively, while bulk dunes had a much lower FeT content
(2.0±0.33 %). Fe-speciation studies reveal that FeS percentage from FeT (FeS/FeT) is the
prevailing Fe mode of occurrence (67±2.4, 69±3.0, 68±2.7 and 73±5.9 % in crusts, paved
sediments, sediments and dunes, respectively), followed by FeD percentage from FeT (FeD/FeT)
(31±2.3, 29±3.0, 30±3.0, 26±5.8 %), and FeA percentage from FeT (FeA/FeT) (1.9±0.55, 1.7±0.56,
1.4±0.55 and 1.0±0.54 %). These results show that FeT is very similar between crusts, paved
sediments and sediments while FeT in dunes is depleted by almost 50 %. Compared to Shi et al.
(2011) samples from northwestern Africa, our sample is depleted in total iron (4.7 % FeT from
Shi et al. (2011)), quite similar in FeS (67 % from Shi et al. (2011)), similar in FeD (33 % from Shi
et al. (2011)) and much higher in FeA (0.43 % from Shi et al. (2011)).
The mean FeT content in the basin is similar in Erg Smar (3.6±0.27 %) and L'Bour (3.2±0.66 %)
compared to high-lands (3.0±0.24 %). The ratio FeA/FeT was slightly higher at Erg Smar (1.9±0.53
%) but similar at L'Bour and high-lands (1.3±0.44 and 1.5±0.47 %, respectively). This is probably
due to the preferential accumulation of exchangeable and nano-Fe-Oxides (FeA) in the low-
lands, where flooding results in red-water ponds and red surfaces after drying. Subsequently,
highly concentrated ionic Fe is trapped in the last stages of ponding, and nano-Fe-oxides may
precipitate during drying. Once the ponded is dried, the crusts of the low-lands tend to have a
reddish patina (see section 3.4). However, a slightly higher mean FeD/FeT of 33±2.4 % is
obtained in the high-lands compared to 31±2.7 and 29±2.4 % at L'Bour and Erg Smar,
respectively. The FeS/FeT mean content is slightly lower at the high-lands (65±2.5 %) compared
to Erg Smar and L'Bour (69±2.6 and 68±2.6 %, respectively).
FeD levels were apportioned between hematite and goethite using XRD proportions. These
results show that in crusts, 0.79±0.66 % of hematite and 0.55±0.67 % of goethite are present, in
paved sediments 0.83±0.51 and 0.64±0.54 %, in sediments 0.73±0.58 and 0.69±0.59 %, and in
dunes 0.20±0.17 % and 0.68±0.24 %.
The proportions of FeD + FeA are higher in crusts, probably due to preferential transport of non-
FeS to the low-lands and the trapping of Fe ions (FeA) by clay adsorption during ponding, and
the formation of nanosized Ferrihydrite ($Fe_{4-5}(OH,O)_{12}$. This readily exchangeable Fe has very low
impact on radiative forcing but a high impact in Fe fertilisation of oceans during dust events
(Gobler et al., 2001), as ionic Fe adsorbed by clays and nano-Fe-oxides are easily released in
water solutions. The correlation of FeS, FeD and FeA with FeT is linear, with coefficients of
determination ($R^2$) reaching 0.96, 0.89 and 0.67 for FeS, FeD and FeA respectively (Figure S4).
Thus, when increasing total Fe content all modes of occurrence of Fe increase, but the increase
is preferentially driven by FeS, while it seems that the basin FeA segregation causes a lower
correlation with FeT.



**3.4 Conceptual model for grain size and mineralogy fractionation in crusts and paved sediments**

According to Bullard et al. (2011) and as previously discussed in this study, heavy rainfall results in the selective deposition of coarser particles from runoff and floodwaters in higher elevations. Conversely, smaller particles enriched in clays, colloidal Fe-oxides (which give the water a reddish hue), and dissolved salts tend to be transported to lower elevations. Figure 7 summarises a conceptual model that outlines the formation of crusts and paved sediment in the study area, with a focus on particle size and mineralogical fractionation.

In the low-lands, floodwaters carrying fine sediments flood extensive flat areas, such as Erg Smar or Iriki lake. Prospero et al. (2002), Bullard et al (2011) and Ginoux et al (2012), among others, have shown that dust emissions originate from relatively small and localised areas where sediments are supplied by floodwaters, and that the occurrence of dust emissions from these areas may be partly due to the occurrence or absence of floodings. During ponding in low-lands, coarser particles deposit first and form a high sand-rich bottom layer of the crust (as described in section 3.2) (Figure 7a & 8a). Subsequently, the clay fraction deposits on top of the bottom layer until total dryness (Figure 7a & 8a) forming a second clay-rich fraction layer in the crust. However, the particle size in crust surfaces is heterogeneous (Figure S5 & S6), which can result in erodible dust-emitting sediment (heterogeneity enhances sandblasting). The finer and more easily exchangeable FeA fraction remains in suspension until the last drying stages on the most superficial layer of the crust, during drying out of the remaining ponds (as described in section 3.3) (Figure 7a & 8a). During this ponding, dissolved Fe ions interact with clays in such a way that they can be adsorbed on clay surfaces according to the ionic composition of the waters (as described in section 3.3) (Echeverría et al., 1998). This typical ion adsorption by clays is higher for montmorillonite than for other clays but the content of montmorillonite is low compared to illite. In this study a high correlation is obtained for FeA and illite contents (Figure S7). Furthermore, crusts contain a higher proportion of hematite(oxide)/goethite(hydroxide) in the FeD, due to the weathering with water during transport and ponding and precipitation of nano-Fe-oxides during drying.

After the pond drying, the continuous heating of the clay rich surface layer causes the hardening of the crust and mud-cracking, giving a 'ceramic-like' compactness to the thick crusts in the low-lands, usually with a reddish colour induced by the Fe-oxides (Figure S5a). Complete drying causes mud cracks due to loss of volume, breaking the crust into polygonal pieces, whose thickness and area depend on the amount of clay deposited. Furthermore, these concave mud-crust pieces resulting from the cracking usually have a grey-colour patch in the middle due to the superficial precipitation of salts, which together with carbonates accumulate by capillarity (see section 3.1.2) (Figure S5b). This capillary ascension and precipitation of salts (the latter being an expansive process) causes sponging and breaking of the surface layers. Thus, a third (top) layer is formed in the crusts of the low-lands, which is very easily eroded by wind because of the spongy structure and enriched in clay and readily exchangeable Fe. In some cases, in Erg Smar, we observed an additional breaking and sponging of the third (upper layer) due to expansive clays. Both the ceramic-like compactness and the cementing of salts give the fine-clay rich crusts in the low-lands a compact pattern with coarser MDPSD compared with the high-lands where ponding is limited and very thin crusts occur. This could explain why the crusts from



the low-lands have finer FDPSDs and coarser MDPSDs compared to the high-lands (see section
3.1.1). Also, wind erosion of the few top millimetres of these crusts may result in dust with higher
contents of clay, Fe-oxide and salts compared to a 15 cm sediment profile.
In the high-lands, washout erosion occurs during rainfall, leading to the formation very thin
crusts in reduced areas. This results in sources of dust made of very thin crusts and fields of
stony surfaces with lower emission rates compared to the low-lands (Bullard et al., 2011). As
illustrated in Figure S6 the surfaces of paved sediments and their thin crusts might resemble
crusts profiles, but with the top section depleted on clay minerals due to preferential erosion
over time, and with a very thin layer (a few micrometres) of clay minerals from the previous
intact formed crust after flooding or running water. The top paved sediments are more compact,
finer and have homogeneous distribution of the particle than crusts, which makes them less
erodible and less likely to emit dust compared to crusts (which have heterogeneous particle size,
see section 3.2).
**4. Conclusions**
This study analysed the particle size and mineralogy of dust-emitting sediments in the region of
Drâa basin in Northern Africa, at the northwestern fringe of the Sahara. The study aimed to
compare these patterns for different types of sediments and their variations across the basin.
The results are consistent with the conceptual models of dust emission sources in desert areas
of Prospero et al. (2002) and Bullard et al. (2011), which predict higher dust emissions in the
low-lands than in the high-lands. The study shows a clear size and mineralogical fractionation
between paleo-sediments and low-land dust-emitting sediments, indicating that collecting
samples of parent paleo-sediments for particle size and mineralogy may not fully represent the
highly emitting dust sources.
Both PSDs and mineralogy are segregated by transport and deposition of sediments during
runoff of water across the basin, and by the precipitation of salts, which causes a sedimentary
fractionation. Coarser particles such as quartz, feldspars, and carbonates (detrital) deposit first
due to friction and gravity and are enriched in high-lands. In contrast, waters reaching the low-
lands are enriched in fine particles (clays), carbonate, salt and Fe ions from partial dissolution of
minerals of the source lands. When these waters are ponded in low-lands, coarser minerals
deposit first, followed by a second layer enriched in clays minerals. Evaporation of the last
ponded water layer causes the deposition of the finest particles and clays enriched in readily
exchangeable ions of Fe. Once dried, the heating of the surfaces by insolation causes
evaporation of interstitial solutions moving towards the surface by capillarity, leading to the
precipitation of salts and secondary carbonates in the upper layer. This expansive process
sponges the surface of the crust, in some cases accelerated by the occurrence of expansive clays,
which might favour dust emission from a top clay-Fe-salts rich micro-layer. Therefore, dust
emission is not only higher but also has a different mineral composition in the low-lands than in
high-lands that is also controlled by the type of sediment.
Our results show that modeling mineral-speciated dust emission requires understanding of the
the mineralogical and size fractionation of accumulated sediments across inland enclosed
basins. Large areas may act as sediment suppliers, while reduced areas may act as dust emitters



with differences in sediment composition. Models that represent mineral-speciated dust
emission and transport should be developed to properly account for these factors.
Our results have also shown that global atlases fail to describe the clay-size fraction of dust-
emitting sediments in the region, overestimating the clay mineral content and underestimating
that of quartz, feldspars, and Fe-oxides. Quartz and feldspars are overestimated and clay
minerals underestimated in the silt-size fractions. Kaolinite-chlorite are not differentiated, while
our study observes major differences. The classical procedure loses salts during fractionation,
and Fe-oxides are detected mainly by color without precision. Our study detects dolomite,
palygorskite, and smectite, and provides more precision for Fe-oxides, with the mode of
occurrence of Fe in different types of samples and locations. However, the study was unable to
obtain a sample below 10 µm without losing salts in the process.
Dust models need global observationally constrained high-resolution mineral maps, which will
soon become available based on high-quality spaceborne spectroscopy measurements
performed from the International Space Station (Figure 1c, Green et al., 2020). A key challenge
of mineral mapping based on spectroscopy for dust emission modeling is to constrain not only
the presence (Figure 1c) but also the abundance of the different surface minerals. The data
gathered and analysed in this study will be used to evaluate these spaceborne retrievals in
forthcoming studies.




















**Acknowledgments**

The field campaign and its associated research, including this work, was primarily funded by the European Research Council under the Horizon 2020 research and innovation programme through the ERC Consolidator Grant FRAGMENT (grant agreement No. 773051) and the AXA Research Fund through the AXA Chair on Sand and Dust Storms at BSC. **CGF** was supported by a PhD fellowship from the Agència de Gestió d'Ajuts Universitaris i de Recerca (AGAUR) grant 2020_FI B 00678. **KK** was funded by the Deutsche Forschungsgemeinschaft (DFG, German Research Foundation) − 264907654; 416816480. **MK** has received funding through the Helmholtz Association's Initiative and Networking Fund (grant agreement no. VH-NG-1533).

We acknowledge the EMIT project, which is supported by the NASA Earth Venture Instrument program, under the Earth Science Division of the Science Mission Directorate. We thank Dr. Santiago Beguería from the National Scientific Council of Spain for facilitating a field site in Zaragoza, Spain, to test our instrumentaltion and field procedures prior to our campaign in Morocco. We thank Paul Ginoux for providing high-resolution global dust source maps, which were very helpful for the identification of the FRAGMENT experimental sites. We thank Prof. Kamal Taj Eddine from Cady Ayyad University, Marrakesh, Morocco for his invaluable support and suggestions for the preparation of the field campaign. We thank Prof. Bethany L. Ehlmann and Dr. Rebecca Greenberger for the help collecting samples, doing infrared in situ spectroscopy and discussion analysis and to PhD. Abigail M. Keebler for discussion analysis. We thank Houssine Dakhamat and the crew of Hotel Chez le Pacha in M'hamid El Ghizlane for their support during the campaign.

**Credit authorship contribution statement**

**CPG-P** proposed and designed the field campaign with contributions of **AA, KK, MK and XQ**. The Campaign was implemented by **CPG-P, AA, CGF, AGR, KK, MK, AP, XQ, CR** and **JYD**. The samples were collected by **CPG-P, AA, AGR, MK and XQ** and analysed by **CB, PC, AGR, CR** and **ZS**. Spectroscopy was analysed by **RNC**. **AGR** performed the visualization and writing of the original draft manuscript and **CPG-P** and **XQ** supervised the work. **CPG-P** and **XQ** re-edited the manuscript and all authors contributed in data discussion, reviewing and manuscript finalization.

**Declaration of competing interest**

Some authors are members of the editorial board of journal ACP. The peer-review process was guided by an independent editor, and the authors have also no other competing interests to declare.

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





**Figure captions**

**Figure 1.** a) Location of the study area (exact location of data measurement "star": 29°49'30"N, 5°52'25"W), near M'Hamid el Ghizlane, into the Drâa basin in S Morocco. Base layer from world imagery of Google Earth Pro v:7.3.6.9345. b) Frequency of ocurrence (%) of dust optical depth above 0.2 in September, October and November between 2003 and 2016 derived from MODIS Deep Blue. c) EMIT scenes emit20220903t082303_o24606_s001_l2a_rfl_b0106_v01 and emit20230206t101334_o03707_s000_l2a_rfl_b0106_v01 at 60 meters per pixel show the diversity of Fe2+ and Fe3+ bearing minerals (left) and the EMIT 8 phyllosilicates, carbonates, and sulphates (right). The mineral maps were produced by tetracorder 5.27c1 (Clark, 2023). There is some mapped mineralogy difference at the scene boundaries, possibly due to the changing viewing geometry, and variation in atmospheric removal between the two scenes. Cirrus clouds in the scene on the right may also be impacting derived mineralogy.

**Figure 2.** Images of samples collected during a field campaign near M'Hamid el Ghizlane, into the Draa Basin, S Morocco.

**Figure 3.** Median minimally and fully dispersed PSDs of crusts, sediments, paved sediments and dunes. (a) MDPSDs and (b) FDPSDs combined from crust, sediment and paved sediment samples; (c) and (d) are MDPSDs and FDPSDs for dune samples; (e) and (f) are MDPSDs and FDPSDs differentiated by type of sample.

**Figure 4**. Boxplot of median particle size diameters in μm including both fully and minimally dispersed analysis (a) for all samples combined excluding dunes and (b) for dune samples only. Also particle size diameter in μm for crusts, sediment and paved sediment for (c) minimally dispersed and (d) fully dispersed results. Means median diameters for each sediment type are shown with crosses.

**Figure 5.** Spatial variation map with crust fully dispersed mean median particle diameter.

**Figure 6.** Mean mineral group content of dune, crust, paved sediment and sediment samples, and also at Erg Smar, L'Bour and High-lands. Solid lines mark the mean content of all the samples (excluding dune samples). The dashed line divides between type and location of the samples.

**Figure 7.** Schematic model of sedimentation and deposition processes in our study site from high-lands to low-lands for a) crusts and for b) paved sediments.

**Figure 8.** Dust emission conceptual model integrating particle size distributions and mineralogy of dust source sediments. a) Refers to the conceptual thickness and particle size distributions along the basin, b) to the particle size distribution and segregation of mineralogy and c) to the dust emission quantity expected depending on the place in the basin.







Figure 1.



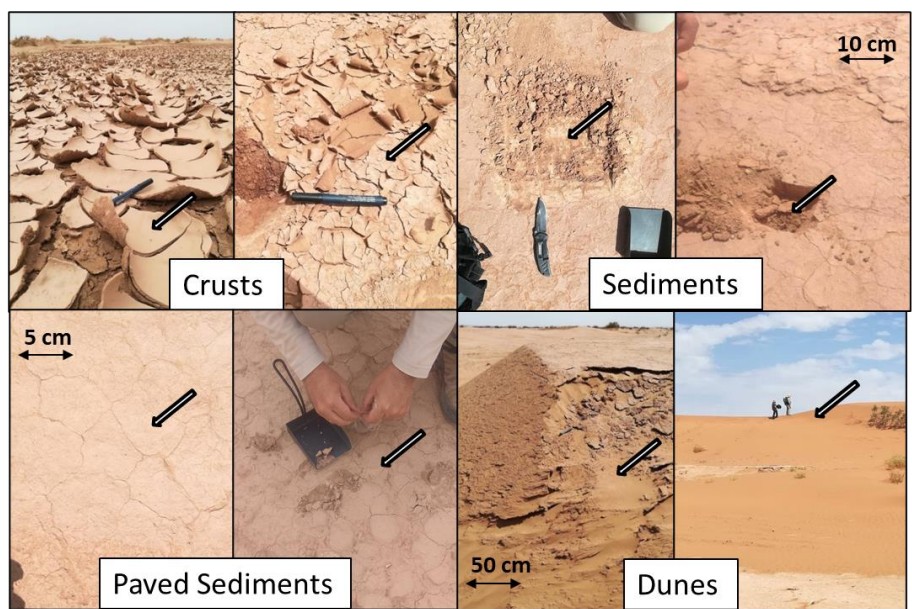

Figure 2.





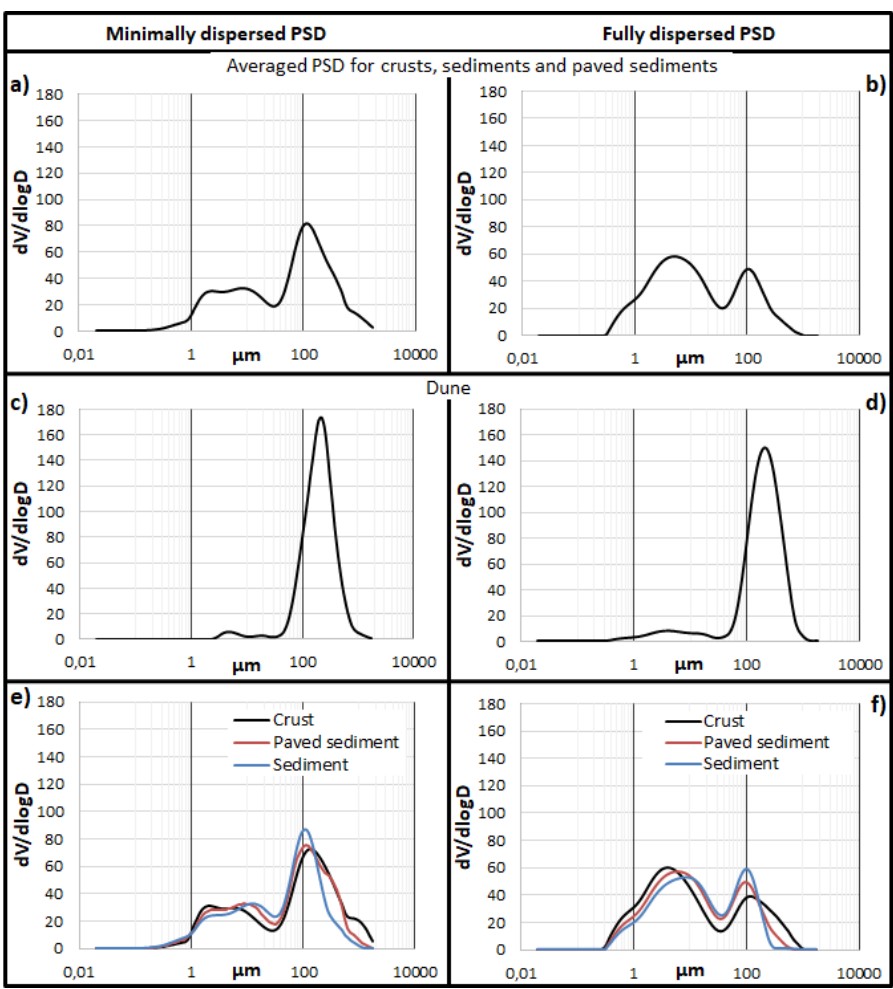

Figure 3.



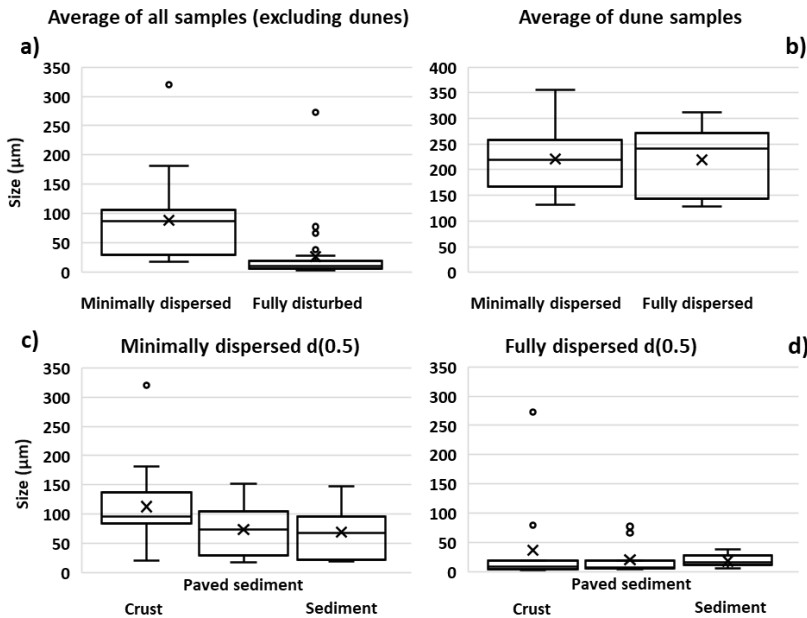

Figure 4.

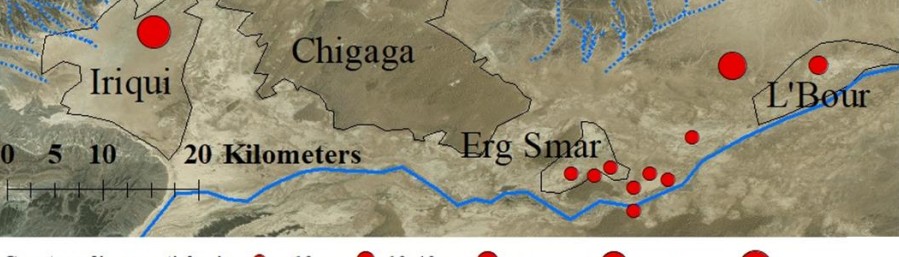

Figure 5.





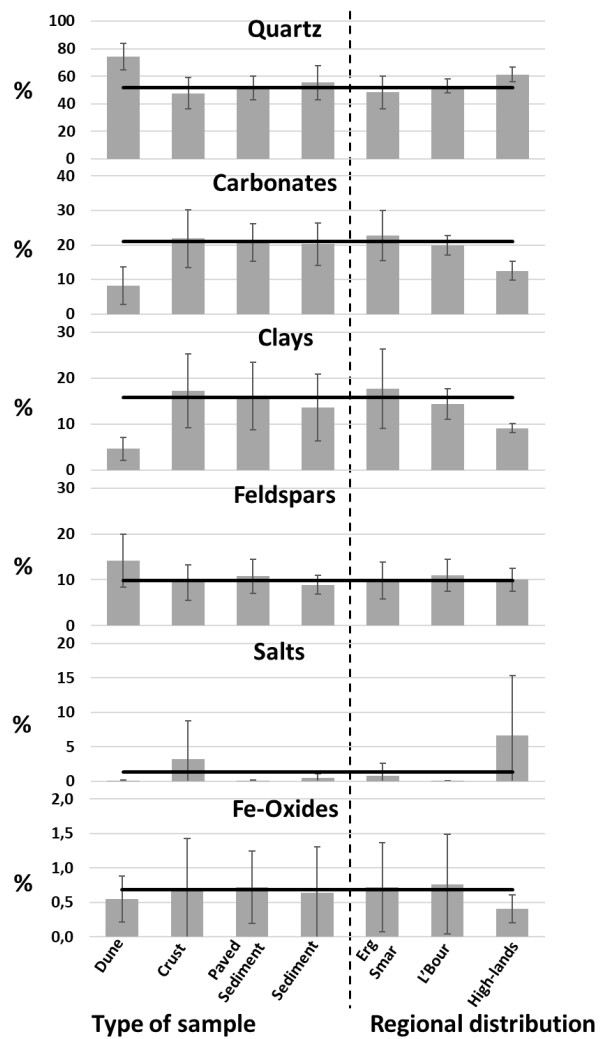

Figure 6.



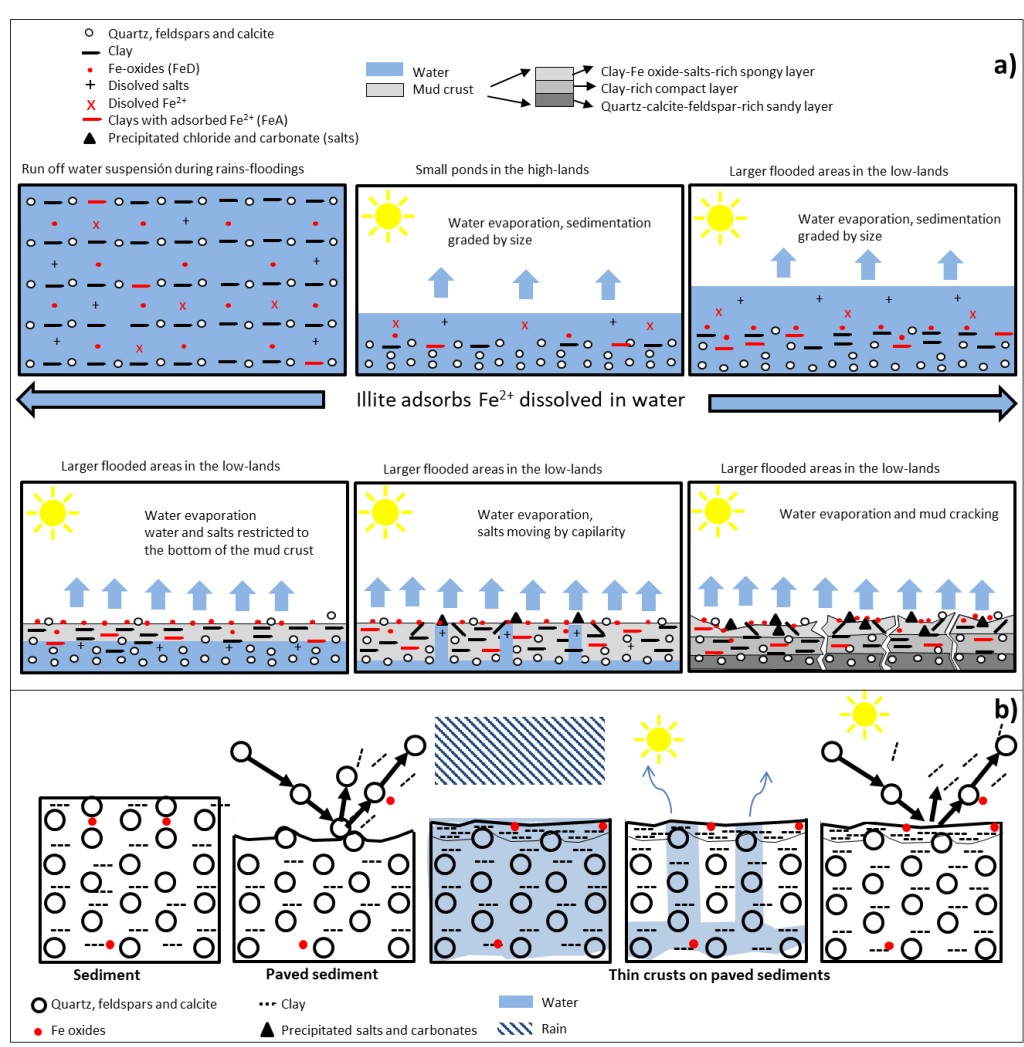

Figure 7.



### a. Macro-scale (basin) size and mineral segregation of sediments

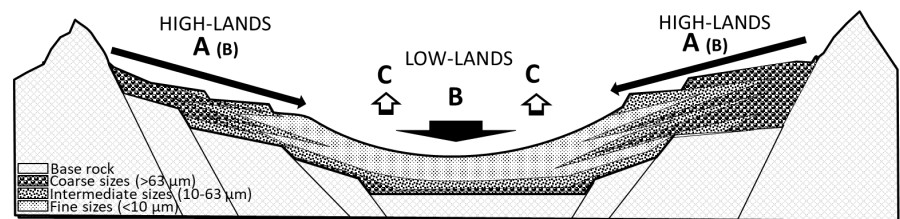

A: Washout, erosion and sporadic flooding with deposition of coarser sediments enriched in quartz and feldspars
B: Flooding and deposition of finer sediments enriched in clays and Fe oxides
C: Evaporation and deposition of fine clays and readily exchangeable Fe oxide, salt crystallization in upper layers

### b. Micro-scale (profiles of deposited sediments) size and mineral segregation

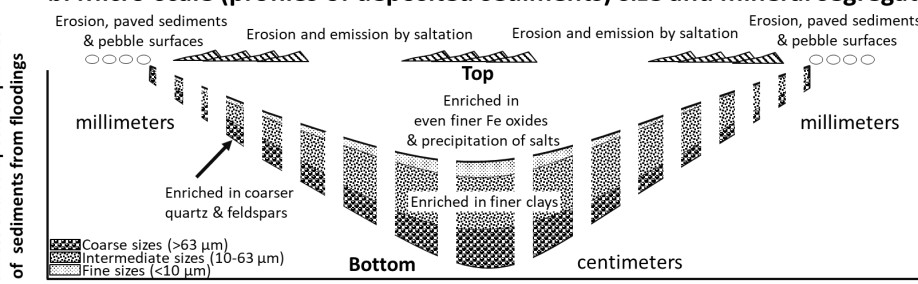

### c. Higher dust emissions (high Fe oxide and clay) in low-lands with thicker & finer deposited sediments

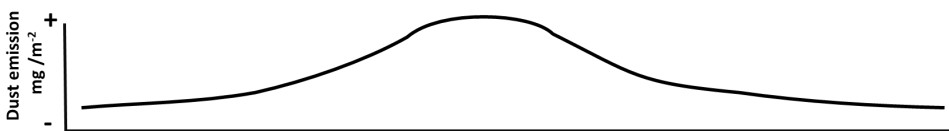

**a+b+c= Emitted dust might be markedly enriched in clays and Fe oxides compared to the parent sediments/soils**

Figure 8.



Table 1. Full range, <63µm and >63 to 2000 µm mean diameter, standard deviation, min., max. and for Minimally dispersed particle size distribution and fully dispersed particle size distribution.

| Surface Type | Location | Nº of samples | MDPSD | | |
|---|---|---|---|---|---|
| | | | Full range | ≤ 63 µm | >63 to 2000 µm |
| | | | Mean of medians ± sd [Min,Max] | | |
| Dunes | Mean | 12 | 221 ± 64 [132,355] | 32 ± 9.3 [20,46] | 252 ± 65 [142,364] |
| | High-Land | 3 | 212 ± 27 [195,243] | 45 ± 1.3 [44,46] | 259 ± 22 [243,284] |
| | Erg Smar | 4 | 286 ± 49 [244,355] | 32 ± 8.1 [25,41] | 295 ± 52 [238,364] |
| | L'Bour | 5 | 174 ± 45 [132,244] | 27 ± 7.4 [20,36] | 214 ± 76 [142,332] |
| Crusts | Mean | 12 | 113 ± 79 [20, 320] | 15 ± 3.7 [7.7,19] | 308 ± 146 [146,635] |
| | High-Land | 3 | 94 ± 5 [89,99] | 18 ± 1.1 [17,19] | 219 ± 28 [187,238] |
| | Erg Smar | 8 | 131 ± 89 [21,320] | 13 ± 3.4 [7.7,17] | 362 ± 151 [193,635] |
| | L'Bour | 1 | 20 ± NA [NA,NA] | 15 ± NA [NA,NA] | 146 ± NA [NA,NA] |
| Paved Sediments | Mean | 11 | 74 ± 48 [19,152] | 17 ± 6.7 [11,33] | 237 ± 71 [146,387] |
| | High-Land | 0 | NA | NA | NA |
| | Erg Smar | 8 | 68 ± 46 [19, 148] | 17 ± 7.0 [11,33] | 240 ± 43 [167,320] |
| | L'Bour | 3 | 90 ± 61 [29,148] | 18 ± 7.1 [13,26] | 230 ± 137 [146,387] |
| Sediments | Mean | 7 | 70 ± 45 [20,147] | 18 ± 5.1 [15,29] | 175 ± 58 [129,302] |
| | High-Land | 1 | 97 ± NA [NA,NA] | 18 ± NA [NA,NA] | 149 ± NA [NA,NA] |
| | Erg Smar | 2 | 115 ± 45 [83,147] | 22 ± 11 [15,29] | 229 ± 104 [155,302] |
| | L'Bour | 4 | 40 ± 23 [20,68] | 17 ± 0.79 [16,17] | 155 ± 21 [129,178] |
| | | | FDPSD | | |
| Dunes | Mean | 12 | 219 ± 70 [128,312] | 24 ± 13 [9.0,46] | 247 ± 72 [145,355] |
| | High-Land | 3 | 250 ± 73 [169,312] | 41 ± 6.8 [33,46] | 290 ± 77 [205,355] |
| | Erg Smar | 4 | 263 ± 32 [239,308] | 20 ± 6.2 [13,25] | 279 ± 33 [238,319] |
| | L'Bour | 5 | 166 ± 61 [128,272] | 16 ± 7.5 [9.0,26] | 195 ± 68 [145,310] |
| Crusts | Mean | 12 | 37 ± 77 [2.7,272] | 9.8 ± 3.6 [3.6,16] | 196 ± 76 [119,389] |
| | High-Land | 3 | 124 ± 132 [20,272] | 13 ± 1.1 [12,14] | 251 ± 121 [162,389] |
| | Erg Smar | 8 | 7 ± 3 [2.7,10] | 7.9 ± 2.5 [3.6,11] | 183 ± 44 [130,236] |
| | L'Bour | 1 | 17 ± NA [NA,NA] | 16 ± NA [NA,NA] | 119 ± NA [NA,NA] |
| Paved Sediments | Mean | 11 | 21 ± 26 [2.3,78] | 13 ± 4.8 [8.2,21] | 157 ± 36 [120,221] |
| | High-Land | 0 | NA | NA | NA |
| | Erg Smar | 8 | 18 ± 24 [5.9,78] | 12 ± 4.6 [8.2,21] | 169 ± 34 [129,221] |
| | L'Bour | 3 | 29 ± 33 [5.3,67] | 14 ± 6.0 [8.3,20] | 122 ± 2.2 [120,124] |
| Sediments | Mean | 7 | 19 ± 11 [5.8,39] | 14 ± 3.9 [7.7,19] | 128 ± 9.6 [117,144] |
| | High-Land | 1 | 12 ± NA [NA,NA] | 9.9 ± NA [NA,NA] | 133 ± NA [NA,NA] |
| | Erg Smar | 2 | 22 ± 23 [5.8,39] | 13 ± 8.1 [7.7,19] | 126 ± 13 [117,135] |
| | L'Bour | 4 | 19 ± 6.3 [13,28] | 15 ± 1.3 [13,17] | 128 ± 11 [122,144] |





Table 2. Mineral results from samples and type of sample. In type of samples, C: Crust, PS: Paved sediment, S: Sediment, D: Dune. In Loc (Location), ES: Erg Smar, LB: L'Bour, HL: High-lands. Sme: Smectite, Mca: Mica/Illite, Kln: Kaolinite, Chl: Chlorite, Plg: Palygorskite, Qtz: Quartz, Cal: Calcite, Dol: Dolomite, Hl: Halite, Gp: Gypsum, Mc: Microcline, Ab: Albite and anorthite, Hem: Hematite, Gt: Goethite. <0.1 indicates below limit of detection.

| | | | Feldspars | | Carbonates | | Clays | | | | | Salts | | Iron Oxides | |
|---|---|---|---|---|---|---|---|---|---|---|---|---|---|---|---|
| Type | Loc | Qtz | Mc | Ab | Cal | Dol | Sme | Mca | Kln | Chl | Plg | Hl | Gp | Hem | Gt |
| C | ES | 55 | 2,6 | 4,8 | 20 | 3,3 | <0.1 | 11 | <0.1 | 1,2 | <0.1 | <0.1 | <0.1 | 1,2 | <0.1 |
| C | ES | 57 | 2,7 | 3,1 | 20 | 3,4 | <0.1 | 5,2 | <0.1 | 0,78 | 0,26 | 7,2 | <0.1 | 0,87 | <0.1 |
| C | ES | 36 | 2,2 | 10 | 21 | 2,7 | <0.1 | 15 | 10,0 | 1,4 | <0.1 | <0.1 | 0,20 | 1,2 | <0.1 |
| C | ES | 32 | 1,7 | 3,3 | 29 | 3,4 | <0.1 | 10 | 17 | 2,2 | 0,20 | <0.1 | <0.1 | 0,24 | 1,3 |
| C | ES | 38 | 3,7 | 4,7 | 18 | 6,2 | <0.1 | 14 | 9,0 | 1,3 | 0,14 | 3,5 | 0,14 | 0,95 | <0.1 |
| C | ES | 50 | 5,5 | 5,5 | 14 | 2,8 | <0.1 | 12 | 7,9 | 1,3 | <0.1 | <0.1 | <0.1 | 0,21 | 0,85 |
| C | LB | 50 | 13 | 5,1 | 12 | 3,6 | <0.1 | 8,1 | 5,7 | 0,46 | <0.1 | <0.1 | <0.1 | 0,92 | <0.1 |
| C | HL | 63 | 6,9 | 6,8 | 12 | 2,2 | <0.1 | 4,5 | 3,9 | 0,19 | <0.1 | <0.1 | <0.1 | 0,11 | 0,40 |
| C | ES | 45 | 3,7 | 3,2 | 26 | 3,2 | <0.1 | 11 | 5,4 | 1,8 | 0,21 | <0.1 | <0.1 | <0.1 | 0,18 |
| C | ES | 30 | 2,6 | 3,4 | 35 | 2,5 | 0,57 | 8,8 | 14 | 1,4 | 1,5 | 0,14 | <0.1 | 0,14 | 0,17 |
| C | HL | 60 | 3,7 | 5,5 | 11 | 0,98 | <0.1 | 5,7 | 3,4 | 0,97 | 0,19 | 8,1 | 0,21 | 0,41 | 0,60 |
| C | HL | 54 | 4,7 | 3,9 | 7,1 | 1,79 | <0.1 | 5,4 | 3,3 | 0,60 | <0.1 | 16 | 2,0 | 0,22 | 0,65 |
| S | ES | 35 | 1,8 | 4,1 | 24 | 5,6 | <0.1 | 17 | 8,3 | 2,2 | <0.1 | 1,1 | 0,19 | 1,1 | <0.1 |
| S | ES | 67 | 6,6 | 5,1 | 10 | 2,1 | <0.1 | 3,0 | 3,6 | 0,64 | <0.1 | 1,1 | <0.1 | 0,42 | 0,23 |
| S | LB | 51 | 4,6 | 7,9 | 15 | 4,2 | <0.1 | 8,9 | 6,6 | 0,89 | <0.1 | <0.1 | <0.1 | <0.1 | 0,82 |
| S | LB | 57 | 2,7 | 7,8 | 16 | 3,8 | <0.1 | 9,6 | 1,9 | 0,49 | <0.1 | <0.1 | <0.1 | 0,93 | <0.1 |
| S | LB | 57 | 3,4 | 5,4 | 18 | 3,2 | <0.1 | 6,5 | 3,5 | 2,1 | <0.1 | <0.1 | <0.1 | 0,33 | 0,60 |
| S | HL | 67 | 3,2 | 5,3 | 13 | 1,7 | 0,13 | 4,3 | 3,2 | 0,20 | 0,20 | <0.1 | <0.1 | <0.1 | 0,90 |
| S | LB | 51 | 3,4 | 5,1 | 21 | 3,3 | <0.1 | 8,5 | 4,5 | 2,2 | 0,15 | <0.1 | <0.1 | 0,66 | 0,50 |
| PS | ES | 44 | 3,0 | 5,7 | 15 | 3,1 | <0.1 | 16 | 11 | 1,5 | <0.1 | <0.1 | <0.1 | 0,35 | 0,64 |
| PS | ES | 44 | 2,2 | 5,4 | 22 | 4,7 | <0.1 | 13 | 6,8 | 0,54 | <0.1 | <0.1 | <0.1 | 1,1 | <0.1 |
| PS | ES | 55 | 2,3 | 5,4 | 24 | 3,6 | <0.1 | 7,8 | 0,84 | 0,28 | 0,17 | <0.1 | <0.1 | 0,98 | <0.1 |
| PS | ES | 40 | 5,3 | 4,7 | 20 | 4,3 | <0.1 | 13 | 10 | 1,1 | <0.1 | <0.1 | 0,29 | 0,77 | 0,23 |
| PS | ES | 67 | 8,8 | 8,7 | 8,9 | 1,8 | <0.1 | 3,1 | 0,38 | 0,30 | <0.1 | <0.1 | <0.1 | 0,30 | 0,29 |
| PS | LB | 48 | 5,5 | 4,0 | 16 | 4,3 | <0.1 | 11 | 8,7 | 1,3 | 0,13 | <0.1 | <0.1 | 1,1 | <0.1 |
| PS | ES | 61 | 3,5 | 6,3 | 12 | 3,6 | <0.1 | 7,9 | 3,6 | 0,78 | 0,16 | <0.1 | <0.1 | 0,41 | 0,33 |
| PS | ES | 46 | 9,1 | 9,0 | 14 | 3,3 | 0,29 | 6,6 | 8,8 | 1,0 | 0,42 | <0.1 | <0.1 | 0,42 | 0,69 |
| PS | ES | 48 | 2,3 | 7,3 | 22 | 3,7 | <0.1 | 8,1 | 6,3 | 1,3 | <0.1 | <0.1 | 0,16 | 0,17 | 1,1 |
| PS | LB | 61 | 4,0 | 8,1 | 13 | 3,3 | <0.1 | 4,3 | 4,2 | 1,2 | <0.1 | <0.1 | <0.1 | 0,16 | 0,64 |
| PS | LB | 51 | 3,6 | 4,4 | 22 | 3,1 | <0.1 | 8,5 | 4,3 | 2,0 | 0,17 | <0.1 | <0.1 | 0,68 | 0,53 |
| D | ES | 80 | 7,1 | 7,0 | 3,1 | 0,69 | <0.1 | 1,4 | <0.1 | <0.1 | <0.1 | <0.1 | <0.1 | <0.1 | 0,38 |
| D | ES | 65 | 14 | 8,3 | 4,1 | 0,90 | <0.1 | 4,8 | 1,9 | <0.1 | <0.1 | <0.1 | <0.1 | <0.1 | 0,37 |
| D | LB | 73 | 7,0 | 11 | 6,6 | 0,31 | <0.1 | 1,2 | 0,19 | 0,23 | <0.1 | <0.1 | <0.1 | 0,11 | 0,32 |
| D | ES | 89 | 2,5 | 3,0 | 2,0 | <0.1 | <0.1 | 0,69 | 1,4 | <0.1 | <0.1 | <0.1 | <0.1 | 0,45 | 0,65 |
| D | ES | 65 | 12 | 5,4 | 11 | 1,3 | 0,13 | 2,7 | 2,1 | 0,38 | <0.1 | <0.1 | <0.1 | 0,12 | 0,32 |
| D | LB | 64 | 5,0 | 6,8 | 10 | 5,0 | <0.1 | 3,1 | 4,3 | 0,50 | <0.1 | <0.1 | <0.1 | <0.1 | 0,61 |
| D | LB | 76 | 4,1 | 6,7 | 6,6 | 0,52 | <0.1 | 2,6 | 2,2 | 0,21 | <0.1 | 0,28 | <0.1 | 0,25 | 0,21 |
| D | LB | 77 | 3,9 | 6,7 | 7,5 | 0,53 | 0,26 | 1,5 | 1,6 | 0,49 | <0.1 | 0,34 | <0.1 | 0,13 | 0,50 |
| D | LB | 57 | 11 | 14 | 7,5 | 1,8 | <0.1 | 3,4 | 3,0 | 0,74 | <0.1 | <0.1 | <0.1 | 0,16 | 0,37 |
| D | HL | 85 | 4,8 | 4,0 | 3,7 | 0,33 | <0.1 | 1,1 | 0,22 | 0,51 | <0.1 | <0.1 | <0.1 | <0.1 | 0,69 |
| D | HL | 82 | 9,2 | 3,3 | 2,8 | 0,17 | 0,15 | 1,2 | 0,67 | <0.1 | <0.1 | <0.1 | <0.1 | 0,20 | 0,29 |
| D | HL | 77 | 6,9 | 7,3 | 4,5 | 0,34 | <0.1 | 1,7 | 1,3 | 0,39 | <0.1 | <0.1 | <0.1 | <0.1 | 0,41 |



Table 3. Mineralogy of specific soils according to Claquin et al. (1999) and Journet et al. (2014) and comparison with the one obtained in this study for six selected samples. Bulk, clay and silt fractions mineralogy (obtained from texture fractionation) and <10 μm and silt (10-63 μm) fractions mineralogy using fully dispersed separation. All content is in mass %.

| | Qtz | Feld | Carbonates | | Clays | | | | | | Salts | | Fe-oxides | |
|---|---|---|---|---|---|---|---|---|---|---|---|---|---|---|
| | | | Cal | Dol | Mca | Chl | Sme | Plg | Kln | Tot.clay | Gp | Hal | Hem | Gt |
| Bulk | 58 | 9.5 | 15 | 2.4 | 6.4 | 1.0 | 0.1 | 0.2 | 3.8 | 11 | 0.2 | 8.1 | 0.5 | 0.5 |
| Clay Ye Claquin | 5 | NA | 6 | NA | 89 | NA | NA | NA | NA | ≈89 | NA | NA | NA | NA |
| Clay Ye Journet | 8 | 3 | 18 | NA | 67 | NA | NA | 1 | 3 | ≈71 | NA | NA | NA | NA |
| Clay Fl Claquin | 12 | NA | 11 | NA | 77 | NA | NA | NA | NA | ≈77 | NA | NA | NA | NA |
| Clay Fl Journet | NA | NA | NA | NA | 98 | NA | NA | 1 | 1 | ≈100 | NA | NA | NA | NA |
| Clay classic Drâa <10μm | 17 | 7.1 | 8.9 | 0.5 | 23 | 9.9 | 1.2 | 1.0 | 22 | 57 | NA | NA | 0.7 | 5.2 |
| FD Drâa | 23 | 4.7 | 19 | 2.4 | 19 | 4.7 | 0.4 | 0.2 | 14 | 38 | NA | NA | 2.2 | 1.8 |
| Silt Ye Claquin | 58 | 31 | 8 | NA | NA | NA | NA | NA | NA | NA | 2 | NA | 1 | NA |
| Silt Ye Journet | 43 | 21 | 20 | NA | 9 | 6 | NA | NA | NA | 15 | NA | NA | 1 | NA |
| Silt Fl Claquin | 30 | 38 | 29 | NA | NA | NA | NA | NA | NA | NA | 2 | NA | NA | NA |
| Silt Fl Journet | 39 | 19 | 12 | NA | 19 | 10 | NA | NA | NA | 29 | NA | NA | 1 | NA |
| Silt classic Drâa | 30 | 8 | 12 | 4.9 | 19 | 6.4 | 0.3 | 0.1 | 13 | 39 | NA | NA | 0.2 | 0.6 |
| Silt FD Drâa | 39 | 8.0 | 23 | 5.0 | 12 | 2.8 | 0.2 | <0.1 | 7.5 | 23 | NA | NA | 1.2 | 0.7 |

Fl: Fluvisol sediment type; Ye: Yermosol; Qtz: Quartz; Feld: Feldspars; Cal: Calcite; Dol: Dolomite; Mca: Mica/illite; Chl: Chlorite; Sme: Smectite; Plg: Palygorskite; Kln: Kaolinite; Gp: Gypsum; Hal: Halite; Hem: Hematite; Gt: Goethite; FD: fully dispersed.