# Peer review of "Variability in grain size, mineralogy, and mode of occurrence of Fe in surface"

_EGUsphere, 2023_

## Referee Comment (RC1)

Dear Author/s

**Egusphere-2023-1120**

The manuscript egusphere-2023-1120 titled (Variability in grain size, mineralogy, and mode of occurrence of Fe in surface sediments of preferential dust-source inland drainage basins: The case of the Lower Drâa Valley, S Morocco) is well written and contains appreciable efforts, we do really enjoy reading the MS and we do encourage to be published meanwhile, the manuscript needs more effort to raise it to the level of acceptance within egusphere. The dust source areas upon definition from the three UN agreements are called (hotspot/s). We were wondering why the authors mentioned sand dunes to be a source (hotspot) for dust? although dunes are mainly composed of sand particles (99%), while mud size fractions are only less than 0.5% of the total weight of all dunes in the world. On the other hand, dunes usually formed in drainage systems which is a source of dust as mentioned by the authors in the abstract and the introduction. So the drainage systems are the source, not the dunes. The introduction contains many old global studies (some are from the last century) and very little from regional (Africa and the Middle East) for comparison and no statement about the importance of this study for Morocco particularly and the region and the methods need some more information mainly within the desert regions. Author/s did not refer to important regional studies as we mentioned. On the other hand, this type of research benefits the surrounding community and humanity not only for research purposes, therefore, some recommendations/solutions should be addressed. Please go through the comments in detail, and hope to see your reply. Adding some regional dust mineralogy and specifications will give support your study.

There are some essential comments authors should take into consideration such as below:

**General remarks:**
- A scientific manuscript should not use (we, I, … and so on) that has been used in the present MS within the abstract, introduction, and methodology sections, we would request to be rewritten.

**Specific remarks:**

- **The title**
  - Too long. To attract citations title should be the fewest words possible, and changed (Grain size) to (particle size) in all the text.
- **Abstract**
  Simplify it as possible and be rewritten (add the number of samples, aim, and importance of this study, and main conclusions and recommendations), most researchers will give focus on your title and abstract, so make it perfect as possible. The advice was given to me by one reviewer a long time ago (always make your fingerprint in your manuscript (figures, tables, text), that if anyone sees it in the street, he knows it belongs to you.
- **Keywords**
  To be changed to (Aeolian; desert, arid land, sand dunes, dust, mineralogy, Morocco)

- **The introduction:**
  Authors need to think outside of Morocco. So introduction should begin with regional studies and then with local studies. Regional references were very little mentioned and some statements are without supportive references. Also, some sentences are mentioned without a reference. Therefore, we suggested to the author to put some supportive references for some statements he mentioned. Such as:
  - Add a reference as follows to show the importance of the study and end with the aim as the last paragraph of the introduction section as follows
    - {Aeolian activities including dust occur predominantly in the desert regions. Aeolian activities may cause direct and indirect adverse effects on fauna, flora, and human health on a regional scale in Iran (Doronzo et al. 2016), Kuwait (Alshemmari et al.2013), and Saudi Arabia (Al-Dousari et al. 2020). It has a socio-economic impact on health (Al-

Dousari et al. 2018), and photovoltaic energy efficiency (Al-Dousari et al. 2019), Therefore the aim of this study is to…..}.
- Delete all references of the last 20 years and add new ones from 2012-2022
- We like when the author wrote in line 160 (For example…. Sahara dust and china dust)
- Remove all (we, I, …..) and replace by (This study show/provide…) line 182
- Replace (we provide ) with (Therefore, the aim of this study is to provide) line 182 and add it to the end of a suggested above sentence

- **Materials and methods**
  o Line 215 (the precipitation) put the average instead of the range (50-800mm), as the range is so variable
  o Do you think 42 samples are representative,
  o A proper sample map is needed for all sampling sites as Fig. 1 is not showing the sampling sites

- **Results and discussion**
  o We think it is essential to add the table below after adding your results, why? Because it gives your study a real comparison and strengthens your discussion section.
  o The authors mentioned Aeolian risk evaluation without referring to sand dunes fluxes in Morocco or the region, therefore, we suggest adding the table below to support this good study with regional data.
  o Add a supportive following table for comparison to justify and support your results

**Table.** Average particle size and mineralogical percentages of deposited dust in Morocco compared to global dust samples.

| Sector | Reference | Size particle % | | Minerals % | | | | |
|---|---|---|---|---|---|---|---|---|
| | | Mud | Sand | Quartz | Feldspars | Carbonates | Clay | Others |
| Morocco | Present study | add | add | add | add | add | add | add |
| Ahwar-Iraq | Doronzo et al. 2016 | 97 | 3 | 13 | 8 | 80 | 0 | 0 |
| Manamah-Bahrain | Al-Dousari et al. 2019 | 87 | 12 | 32 | 10 | 41 | 3 | 15 |
| Walameen-south Saudi | Al-Dousari et al. 2020 | 61 | 40 | 62 | 24 | 13 | 1 | 0 |
| Ain-Emirates | Al-Dousari et al. 2018 | 4 | 97 | 26 | 20 | 52 | 1 | 0 |
| Dubai-Emirates | Subramaniam et al. 2015 | 82 | 17 | 21 | 6 | 45 | 0 | 27 |
| Amman-Jordan | Alshemmari et al.2013 | 70 | 30 | 21 | 4 | 68 | 0 | 7 |
| Tripoli-Libya | Al-Ghadban et al. 1999 | 81 | 20 | 64 | 5 | 27 | 4 | 0 |
| Cartagena-Colombia | Doronzo et al. 2016 | 90 | 10 | 66 | 33 | 0 | 0 | 1 |
| Cairo-Egypt | Al-Dousari et al. 2020 | 90 | 10 | 51 | 15 | 34 | 0 | 0 |
| Bald Hill-Australia | Cattle et al. 2002 | 90 | 9 | 57 | 21 | 0 | 14 | 7 |
| Average | | 75 | 25 | 41 | 14 | 38 | 2 | 5 |

- **Conclusion**
  o Make it much shorter and add a sentence about your recommendations and what is your future / upcoming studies after this research. For example, the effect of native vegetation in reducing Aeolian dust (mobile sand and dust) in the region. Therefore, we suggest adding {Native plants and green belts have also contributed to the reduction in the annual rates of mobile sand by 94 and 95.3%, and dust by 64.5 and 68.4%, respectively (Al-Dousari et al. 2020)}
  o some recommendations/solutions should be addressed. Please go through the comments in detail, and hope to see your reply. Adding some regional dust mineralogy and specifications will give support your study.

Supportive references suggested to be added as mentioned in the comments:
**Suggested references**

Cattle, S.R., McTainsh, G.H., Wagner, S., 2002. Aeolian dust contribution to soil of the Namoi Valley, northern NSW, Australia. Catena 47, 245-264.

Al-Ghadban, et al. (1999). Preliminary assessment of the impact of draining of Iraqi marshes on Kuwait's northern marine environment. parti. physical manipulation. *Water science and technology*, *40*(7), 75-87. https://doi.org/10.1016/S0273-1223(99)00586-7

Alshemmari, et al. (2013). Mineralogical characteristics of surface sediment in Sulaibikhat Bay, Kuwait. *Kuwait Journal of Science*, *40*(2).

Doronzo, et al. (2016). Preface to the Dust Topical Collection. *Arab J Geosci* **9,** 468 (2016). https://doi.org/10.1007/s12517-016-2504-9

Subramaniam, et al. (2015). Probability distribution and extreme value analysis of total suspended particulate matter in Kuwait. *Arabian Journal of Geosciences*, *8*(12), 11329-11344. https://doi.org/10.1007/s12517-015-2008-z

Al-Dousari, A.M., Ibrahim, M.I., Al-Dousari, N. *et al.* (2018). Pollen in aeolian dust with relation to allergy and asthma in Kuwait. *Aerobiologia* **34,** 325–336. https://doi.org/10.1007/s10453-018-9516-8

Al-Dousari, et al. (2019). Off-road vehicle tracks and grazing points in relation to soil compaction and land degradation. *Earth systems and environment*, *3*(3), 471-482. https://doi.org/10.1007/s41748-019-00115-y

Al-Dousari, A., Ramadan, A., Al-Qattan, A., Al-Ateeqi, S., Dashti, H., Ahmed, M., Al-Dousari, N., Al-Hashash, N. and Othman, A., 2020. Cost and effect of native vegetation change on aeolian sand, dust, microclimate and sustainable energy in Kuwait. *Journal of Taibah University for Science*, *14*(1), pp.628-639. https://doi.org/10.1080/16583655.2020.1761662

---

## Referee Comment (RC2)

[referee-annotated manuscript omitted]

---

## Author Response (AR1)

**REPLY TO QUERIES RAISED BY PROF. AL-DOUSARI TO THE MAUSCRIPT EGUSPHERE-2023-1120**

Dear Prof. Al-Dousari,

Thanks a lot for your comments on our manuscript. Please find below our answers to your comments and an explanation of the changes in the revised version of the manuscript (text in red).

With kind regards

The authors

**GENERAL COMMENTS**

**Prof Al-Dousari:** 'The manuscript egusphere-2023-1120 titled (Variability in grain size, mineralogy, and mode of occurrence of Fe in surface sediments of preferential dust-source inland drainage basins: The case of the Lower Drâa Valley, S Morocco) is well written and contains appreciable efforts, we do really enjoy reading the MS and we do encourage to be published meanwhile, the manuscript needs more effort to raise it to the level of acceptance within Egusphere.

Reply: Many thanks for your positive comments.

**Prof Al-Dousari:** 'The dust source areas upon definition from the three UN agreements are called (hotspot/s). We were wondering why the authors mentioned sand dunes to be a source (hotspot) for dust? although dunes are mainly composed of sand particles (99%), while mud size fractions are only less than 0.5% of the total weight of all dunes in the world. On the other hand, dunes usually formed in drainage systems which is a source of dust as mentioned by the authors in the abstract and the introduction. So, the drainage systems are the source, not the dunes.'

Reply: The most general, inclusive, and common term to refer to the locations where dust is emitted in the scientific literature is "dust source". Following your recommendation, we also use the term hotspot as identified by Gillette (1999) to refer to localized, persistent areas of intense dust production within an overall landscape which generally does not emit dust.

Concerning the sand dunes as sources of dust, we were probably not clear enough in the text and this caused confusion. We do agree that dunes are not dust emission hotspots. That said, dust can still be emitted moderately from sand dunes. We refer to the extensive work by Bullard et al. (2011), where sand deposits are discussed and classified into sand sheets and aeolian sand dunes. While dust emission is rather limited in sand sheets, there can be moderate contributions to dust emission from aeolian sand dust dunes depending on type, activity level, and paleoenvironmental history. For example, large, stable or old dunes may accumulate fine material within the dune structure due to weathering or, in semi-arid regions, through in-wash of finer particles by precipitation (Bullard et al. (2011).

In conclusion, we do agree with the reviewer in that dunes are not dust emission hotspots. In the study area, dust is emitted mostly from the mud lands (containing most of the fine particles that emit dust) by saltation and sandblasting with sand provided by surrounding dunes under moderate to strong wind conditions. This is reflected in the introduction section.

**Prof Al-Dousari:** 'The introduction contains many old global studies (some are from the last century) and very little from regional (Africa and the Middle East) for comparison and no statement about the importance of this study for Morocco particularly and the region and the methods need some more information mainly within the desert regions. Author/s did not refer to important regional studies as we mentioned.'

Reply: We have added more regional and updated references from Africa and Middle East and we compared our results with the ones from these studies. The revised manuscript includes the list of references suggested by the reviewer and a new table was added following your suggestions.

**Prof Al-Dousari:** 'On the other hand, this type of research benefits the surrounding community and humanity not only for research purposes, therefore, some recommendations/solutions should be addressed.'

Reply: As explained in the abstract and introduction our research study focuses on understanding the variability of sediment properties across a typical dust-source basin. We

provide some fundamental understanding of these aspects and at the same time our research has practical implications. For example, our data can help constraining high-resolution mineralogical maps for mineral-speciated dust modelling within climate models that are used to study climate change. Our study is not focused on understanding for example how to prevent aeolian erosion and therefore we cannot (and should not) provide recommendations or solutions on the issue because these would not be based on results and/or evidence presented in our study. Per your request we have pointed towards future potential studies in the region that could address those aspects in the conclusions section.

**Prof Al-Dousari:** 'Please go through the comments in detail, and hope to see your reply. Adding some regional dust mineralogy and specifications will give support your study.'
Reply: We have updated the manuscript accordingly.

**SPECIFIC COMMENTS**

1.  **Writing:** A scientific manuscript should not use (we, I, … and so on) that has been used in the present MS within the abstract, introduction, and methodology sections, we would request to be rewritten.
Reply: Done

2.  **Tittle**: Too long. To attract citations title should be the fewest words possible, and changed (Grain size) to (particle size) in all the text.
Reply: We changed grain size by particle size as suggested. We also admit the title is a bit long. We reduced the word count by 5 words, while keeping essential keywords in the title that in our opinion can attract a broad readership.

3.  **Abstract:** Simplify it as possible and be rewritten (add the number of samples, aim, and importance of this study, and main conclusions and recommendations), most researchers will give focus on your title and abstract, so make it perfect as possible. The advice was given to me by one reviewer a long time ago (always make your fingerprint in your manuscript (figures, tables, text), that if anyone sees it in the street, he knows it belongs to you.
Reply: We have improved the abstract, which includes number of samples, aim, and importance of this study, and main conclusions.

4.  **Keywords:** To be changed to (Aeolian; desert, arid land, sand dunes, dust, mineralogy, Morocco
Reply: We have changed the keywords following most of your recommendations.

**5. Introduction**

5.1.    Authors need to think outside of Morocco. So introduction should begin with regional studies and then with local studies. Regional references were very little mentioned and some statements are without supportive references. Also, some sentences are mentioned without a reference. Therefore, we suggested to the author to put some supportive references for some statements he mentioned. Such as: o Add a reference as follows to show the importance of the study and end with the aim as the last paragraph of the introduction section as follows.
Reply: We added the references you suggested and comments to the mineralogy from other source regions in the introductory sections.

5.2.    Aeolian activities including dust occur predominantly in the desert regions. Aeolian activities may cause direct and indirect adverse effects on fauna, flora, and human health on a regional scale in Iran (Doronzo et al. 2016), Kuwait (Alshemmari et al.2013), and Saudi Arabia

(Al-Dousari et al. 2020). It has a socio-economic impact on health (AlDousari et al. 2018), and photovoltaic energy efficiency (Al-Dousari et al. 2019), Therefore the aim of this study is to…..}.
Reply: We added the references you suggested in this section.

5.3.     Delete all references of the last 20 years and add new ones from 2012-2022.
Reply: We added the new references suggested while keeping only some important older references.

5.4.     We like when the author wrote in line 160 (For example…. Sahara dust and china dust).
Reply: Thanks for this. We also added comparison with Middle East based on your reference.

5.5.     Remove all (we, I, …..) and replace by (This study show/provide…) line 18.
Reply: Done.

5.6.     Replace (we provide ) with (Therefore, the aim of this study is to provide) line 182 and add it to the end of a suggested above sentence.
Reply: Done

**6. Materials and methods**
6.1.     Line 215 (the precipitation) put the average instead of the range (50-800mm), as the range is so variable
Reply: An annual average (80 mm) has been added to replace the range.
6.2.     Do you think 42 samples are representative
Reply: These were considered representative because the study focuses on sediments (not deposited dust) and one basin. We added this text in the methodology.

6.3.     A proper sample map is needed for all sampling sites as Fig. 1 is not showing the sampling sites.
Reply: The map you requested was added.

**7. Results and discussion**
We think it is essential to add the table below after adding your results, why? Because it gives your study a real comparison and strengthens your discussion section. The authors mentioned Aeolian risk evaluation without referring to sand dunes fluxes in Morocco or the region, therefore, we suggest adding the table below to support this good study with regional data. Add a supportive following table for comparison to justify and support your results.

Reply: To fulfil your request, the comparison table you suggested was included in the supplemental material. While interesting, the data you suggest is deposition data, but in this paper we discuss analysis of sediments collected from the surface. This is the reason the table was placed in the supplemental material. We added this comment in the results and discussion: 'Table S1 compares the silt+clay and sand proportions and the mineral contents of the crusts from this study in Morocco with those from deposited dust in different arid regions of the world. The FD-PSD data from this study evidences that 72% of the particles in the crusts fall in the clay+silt fraction (<63 µm), while 28% in the sand size-range. This is close to the average value (74 and 26%, respectively) calculated from the existing studies on deposited dust.  Concerning the mineralogy, the crusts of this study are enriched in clays and depleted in carbonate minerals and feldspars compared with the average of the mineralogy of deposited dust shown in Table S1.'

**8. Conclusion**

8.1.	Make it much shorter and add a sentence about your recommendations and what is your future / upcoming studies after this research. For example, the effect of native vegetation in reducing Aeolian dust (mobile sand and dust) in the region. Therefore, we suggest adding: 'Native plants and green belts have also contributed to the reduction in the annual rates of mobile sand by 94 and 95.3%, and dust by 64.5 and 68.4%, respectively (Al-Dousari et al. 2020)' o some recommendations/solutions should be addressed.

Reply:  We revised the conclusions section and added this text 'The large dam built in the Drâa River has caused the drying of this part of the basin, a reduction of vegetation and probably increased dust emissions. The region exemplifies how anthropogenic activities can promote wind erosion and represents a unique location for research on the topic. Future studies may indeed explore many other aspects related to sedimentology, mineralogy, wind erosion, dust emission and anthropogenic impacts, including the study of the introduction of native plants and green belts to reduce wind erosion as has already been done in other regions (Al-Dousari et al. 2020).'

8.2.	Supportive references suggested to be added as mentioned in the comments:
Reply: All these references you suggested were added.

- Cattle, S.R., McTainsh, G.H., Wagner, S., 2002. Aeolian dust contribution to soil of the Namoi Valley, northern NSW, Australia. Catena 47, 245-264.
- Al-Ghadban, et al. (1999). Preliminary assessment of the impact of draining of Iraqi marshes on Kuwait's northern marine environment. parti. physical manipulation. Water science and technology, 40(7), 75-87. https://doi.org/10.1016/S0273-1223(99)00586-7
- Alshemmari, et al. (2013). Mineralogical characteristics of surface sediment in Sulaibikhat Bay, Kuwait. Kuwait Journal of Science, 40(2).
- Doronzo, et al. (2016). Preface to the Dust Topical Collection. Arab J Geosci 9, 468 (2016). https://doi.org/10.1007/s12517-016-2504-9
- Subramaniam, et al. (2015). Probability distribution and extreme value analysis of total suspended particulate matter in Kuwait. Arabian Journal of Geosciences, 8(12), 11329-11344. https://doi.org/10.1007/s12517-015-2008-z
- Al-Dousari, A.M., Ibrahim, M.I., Al-Dousari, N. et al. (2018). Pollen in aeolian dust with relation to allergy and asthma in Kuwait. Aerobiologia 34, 325–336. https://doi.org/10.1007/s10453-018-9516-8
- Al-Dousari, et al. (2019). Off-road vehicle tracks and grazing points in relation to soil compaction and land degradation. Earth systems and environment, 3(3), 471-482. https://doi.org/10.1007/s41748-019-00115-y
- Al-Dousari, A., Ramadan, A., Al-Qattan, A., Al-Ateeqi, S., Dashti, H., Ahmed, M., Al-Dousari, N., Al-Hashash, N. and Othman, A., 2020. Cost and effect of native vegetation change on aeolian sand, dust, microclimate and sustainable energy in Kuwait. Journal of Taibah University for Science, 14(1), pp.628-639. https://doi.org/10.1080/16583655.2020.1761662

References cited in the response:
Bullard, J. E., S. P. Harrison, M. C. Baddock, N. Drake, T. E. Gill, G. McTainsh, and Y. Sun (2011), Preferential dust sources: A geomorphological classification designed for use in global dust-cycle models, J. Geophys. Res., 116, F04034, doi:10.1029/2011JF00

**REPLY TO QUERIES RAISED BY REFEREE #2 TO THE MAUSCRIPT EGUSPHERE-2023-1120**

Dear referee,

Thanks a lot for your positive comments on our manuscript. We have implemented the suggested changes and additions, which improved the quality of the manuscript. We are replying your major queries and describing below how we have implemented these (text in red). For minor suggestions we address you to the revised manuscript.

**GENERAL COMMENTS**

**Referee#2:** 'The manuscript by Gonzalez-Romero et al. provides a comprehensive description and analysis of in-situ measurements of dust size distribution and mineralogy taken in Morocco. It is rare to see published such detailed analysis of soil properties crucial to understand dust impacts on climate and air quality. With all their instruments, they are able to differentiate and quantify the different forms of Fe, which plays a crucial role in dust interactions with radiation in the atmosphere or snow albedo and ocean biogeochemistry. By proceeding with multiple samplings, their results are based on robust statistics. The authors further provide a theory to explain the processes involved in sorting vertically the minerals within the low-level basin. This is an excellent paper well written without superfluous details or incomprehensible jargon (despite many different instruments). The figures are very good and informative. I have minimal comments that I added as sticky notes in the uploaded pdf file. I recommend publication of the present manuscript with minor revision'

Reply: Many thanks for your positive comments. We have updated the manuscript according to your suggestions. Below we only select the reply to major comments you supplied as sticky notes.

**SPECIFIC COMMENTS**

**REFEREE#2: ABSTRACT:** 'Your dataset is much more than providing the distribution of minerals. You are offering a unique dataset characterizing not only a dust source hot spots but also its surroundings, which is important to better understand their respective impacts on climate and air quality, and also their interactions.'
Reply: Thanks a lot for this comment. This is now reflected in the abstract of the revised version of the manuscript.

**REFEREE#2: INTRO-R126:** Modelling efforts have mostly focused on the representation of dust sources and emission (Kok et al., 2021) and the characterization of dust sources is one of the crucial aspects for representing dust mobilisation in models: 'Absolutely not! There are some work focusing on dust sources, but most are about impacts.'
Reply: Thanks. We replaced the text you selected to delete by this one: 'The characterization of dust sources and hotspots is one of the crucial aspects for representing dust mobilisation in models.'

**REFEREE#2: INTRO-R128:** Traditionally, models used aridity as a 128 criterion to identify potential dust sources (Tegen and Fung, 1994). 'False. Vegetation cover is the main criterion.'

Reply: Thanks. We were erroneously using the term aridity as analogous to a lack of vegetation. It now reads "Initially, models used vegetation cover as a criterion to identify potential dust sources (e.g. Tegen and Fung, 1994)."

**REFEREE#2: INTRO-R129:** Satellite retrievals subsequently showed that the most prolific sources occupy a small fraction of arid regions (Prospero et al., 2002; Ginoux et al., 2012). 'Where did you get that in Prospero 2002 and Ginoux 2012?'

Reply: We changed the sentence to: 'Satellite retrievals subsequently showed that the most prolific sources occupy a smaller fraction of arid regions (Prospero et al., 2002; Ginoux et al., 2012). These so-called hotspots or "preferential sources" are found within enclosed basins, where easily eroded soil particles accumulate after fluvial erosion and transport from the surrounding high-lands.'

**REFEREE#2: INTRO-R139:** A more fundamental problem is that the models assume homogene soil properties everywhere while you are going to show their important heterogeneity.

Reply: We added: Models assume relatively homogeneous soil properties due to the lack of data, while there can be significant heterogeneity.

**REFEREE#2: INTRO-R158:** There is a recent paper showing that cloud pH is controlled in great part by calcite from dust (Grider, A., Ponette-González, A. and Heindel, R., 2023. Calcium and ammonium now control the pH of wet and bulk deposition in Ohio, US. Atmospheric Environment, p.119986.). Also, Calcium is controlling heterogeneous reactions of acids on the surface of dust which ultimately affect $O_3$ production (Bauer et al., 2004; Paulot et al., 2016).

Reply: We added: 'Recent studies have shown that cloud pH is controlled in great part by calcite from dust (Grider et al., 2023). Furthermore, Ca is controlling heterogeneous reactions of acids on the surface of dust, which ultimately affect $O_3$ production (Bauer et al., 2004; Paulot et al., 2016).' And references were added.

**REFEREE#2: INTRO-R234:** Would you clarify the difference between paved sediments and crusts is the period of formation. The former dates from thousands of years ago while the latter was formed recently?

Reply: Thanks. Yes, it is as you described. We added: 'The difference between paved sediments and crusts is mostly the period of formation. The former can date to up to thousands of years ago, while the latter was formed recently. However, crust might have finer sediments because these are formed by ponding.'

Additional changes in the revised version of the manuscript: we updated Figure 1 to use more recent EMIT scenes of the region and an updated version of Tetracorder.